# Silk-enabled conformal intraventricular interfaces for minimally invasive neural recordings

Jizhi Liang [1,2,10], Xiner Wang[1,2,10], Zhaohan Chen[3], Xiaoling Wei [2,4], Liuyang Sun [1,2], Keyin Liu[2,4], Zhifeng Shi[5], Tiger H. Tao [1,2,3,4,6,7,8,9] & Zhitao Zhou [2,4]

Flexible neural interfaces capable of monitoring subcortical neuronal activity facilitate the study of deep brain neural circuits and their interactions with the cortex. However, there exists a paucity of translational tools for interfacing subcortical nuclei surfaces within the intraventricular cerebrospinal fluid. Here, we developed a flexible and conformal intraventricular interface (IVI) featuring a deformable microelectrode array paired with a silk scaffold. The IVI can be minimally invasively implanted into the lateral ventricles with the assistance of commonly used clinical catheters, self-unfolding in the cerebrospinal fluid environment to conformally attach to the surfaces of periventricular neural structures, and capturing high-quality signals by virtue of the microelectrode's in-plane shielding. In parkinsonian ewes, the IVI detects deep brain abnormalities and achieves stable, biocompatible in vivo recordings for four weeks. This platform enables chronic monitoring and circuit analysis of healthy and diseased deep brain regions, facilitating studies of neural circuits between periventricular surface neurons and distant brain areas.

The deep brain, comprising subcortical structures such as thalamus, basal ganglia, limbic system, brainstem, and cerebellum, is instrumental in various cognitive, affective, social and essential life functions[1]. Abnormalities in the periventricular structure and function of deep brain regions have been observed in various neurological and psychiatric disorders such as Parkinson's disease (PD), Alzheimer's disease, depression, and autism[2–7]. The neural interface technology for precise detection of neural activity in the deep brain offers opportunities for diagnosis and treatment of deep-brain diseases, as well as

neuroscience research. However, most current research on neural interfaces relies on technologies such as electrocorticography electrode arrays[8–11], Utah arrays[12,13], and intracortical flexible neural probes[14,15]. These are primarily suitable for recording neural signals from the cortical surface, the superficial layers of the gray matter, or the deep nuclei of rodents[16]. Stereo-electroencephalography (SEEG)[17–19] and Neuropixels[20,21] electrodes can be directly implanted into deep brain regions due to their inherent rigidity, but their mechanical mismatch with brain tissue and the method of

[1]2020 X-Lab, Shanghai Institute of Microsystem and Information Technology, Chinese Academy of Sciences, Shanghai 200050, China. [2]School of Graduate Study, University of Chinese Academy of Sciences, Beijing 100049, China. [3]Neuroxess Co., Ltd, Shanghai 200023, China. [4]State Key Laboratory of Transducer Technology, Shanghai Institute of Microsystem and Information Technology, Chinese Academy of Sciences, Shanghai 200050, China. [5]Department of Neurosurgery, Huashan Hospital of Fudan University, Shanghai 200040, China. [6]Center of Materials Science and Optoelectronics Engineering, University of Chinese Academy of Sciences, Beijing 100049, China. [7]Center for Excellence in Brain Science and Intelligence Technology, Chinese Academy of Sciences, Shanghai 200031, China. [8]Guangdong Institute of Intelligence Science and Technology; Hengqin, Zhuhai, Guangdong 519031, China. [9]Tianqiao and Chrissy Chen Institute for Translational Research, Shanghai, China. [10]These authors contributed equally: Jizhi Liang, Xiner Wang. ✉e-mail: tiger@mail.sim.ac.cn; ztzhou@mail.sim.ac.cn

implantation can cause invasive damage to the target nuclei being monitored[22]. Additionally, their axially distributed electrode sites struggle to cover larger areas of the deep brain. Flexible and large-area deep-brain neural interfaces are instrumental for studying the neural circuits in deep-brain regions and their interactions with the cortex[23].

The flexible planar electrode arrays with wide coverage currently lack effective minimally invasive implantation techniques for targeted monitoring in deep brain regions[24,25]. Hence, there is a pressing need for the development of flexible planar neural interface technologies that are compatible with minimally invasive neurosurgical implantation procedures. The principal obstacle lies in achieving a conformal attachment of neural electrodes to the surfaces of periventricular neural structures in situ, amidst the dynamic milieu of cerebrospinal fluid (CSF)[26,27]. This challenge is complicated by the necessity to ensure compatibility with existing minimally invasive neurosurgical practices and to maintain high-quality electrophysiological monitoring in complex noise environments[28]. Therefore, establishing a large-scale, stable neural interface with subcortical nuclei, particularly those periventricular, without compromising the integrity of targeted deep brain tissues, remains a challenging endeavor.

This work aims to develop a flexible, conformal, and chronically implantable intraventricular interface (IVI) compatible with minimally invasive neurosurgical techniques for investigating neural activity on the inner surface of the brain—namely, the CSF-facing surfaces of deep brain structures—without necessitating additional surgical trajectories or inflicting damage on the targeted nuclei. By integrating the deformability of flexible microelectrodes with the shape memory properties of a silk scaffold, the IVI can self-unfold and conformally adhere to the periventricular nuclei within the dynamic CSF environment of the brain ventricles. We conducted a series of in vitro and in vivo electrophysiological experiments and biocompatibility assessments, and validated its clinical translational utility in monitoring intraventricular neural activity through both intraoperative and chronic in vivo experiments on Parkinsonian sheep models[29,30]. The results demonstrate that the IVI is capable of performing intraoperative recordings from the caudate nucleus surface in Parkinsonian sheep to monitor the electrophysiological effects of levodopa treatment, accompanied by high-accuracy discrimination of neural activity across multiple microelectrode sites. Furthermore, it enables chronic recordings of periventricular neural activity for a four-week period, allowing dynamic tracking of disease progression in a chronic Parkinsonian sheep model.

## Results

### Overview of the minimally invasive IVI technology

The silk-enabled IVI is minimally invasively implanted in the lateral ventricles and utilizes the shape memory silk scaffold to self-unfold and interface with the deep-brain neural tissues surrounding the lateral ventricles in the CSF milieu. The flexible deformable microelectrode array (dMEA) ensures conformal contact between the IVI and the subcortical nuclei within deep brain regions. Figure 1a illustrates the establishment of a neural interface with the surface of the caudate nucleus head using silk-enabled minimally invasive IVI technology. The dMEA was fabricated using specialized microelectromechanical systems (MEMS) fabrication methods, compared with those previously reported flexible electrodes[31] (see methods and Supplementary Fig. 1 for the detailed manufacturing process). The microelectrodes are composed of a dual-metal layer framework, sandwiched between three layers of polyimide (PI) for insulation and encapsulation with an overall thickness of 14 μm. Metal layer windows were distributed on both the top and bottom surfaces of the device through PI patterning, and interlayer electrical pathways of the corresponding microelectrode sites and pads were accomplished through metal-sputtered vias through the intermediate PI layer (Fig. 1b).

To assess and refine the layout of the IVIs, we employed mechanical finite element simulations to evaluate the strain distribution on the dMEAs when they were attached to curved surfaces with varying curvatures, and the results indicate that the metal traces remain unyielding. We conducted conformal attachment empirical validation corresponding to the simulations using homemade poly-dimethylsiloxane (PDMS) molds. As shown in the top panel of Fig. 1c, the flexible dMEAs demonstrate excellent conformal attachment capability to curved surfaces with curvature radii of 6 mm, 8 mm, and 10 mm, respectively. These molds were also employed to assess the impact of external cyclic forces on IVI electrical performance in the follow-up experiments. We customized a 40-cm-long Flexible Printed Circuit Board (FPC) with a flat connector adapter board to improve operability during surgery. After being soldered to the FPC, the dMEA was integrated with the silk scaffold and assembled into a medical catheter, forming a complete implantable device designed for minimally invasive implantation (see methods and Supplementary Fig. 2). To ensure stable long-term recordings, a skull-mounted base fabricated from high-strength 3D-printed nylon was designed to protect the backend of the IVI (Fig. 1d). The interface device including the dMEA and the silk scaffold was designed to be able to conformally attach to the implant target such as the caudate nucleus head (Fig. 1e).

Minimally invasive neurosurgical procedures (such as external ventricular drainage and endoscopic third ventriculostomy) involving deep brain regions are quite sophisticated[32,33], and establishing a neural interface in conjunction with these procedures necessitates addressing the extra challenge of achieving stable conformal contact with subcortical nuclei in the CSF environment. We proposed a silk-enabled strategy for minimally invasive implantation and self-unfolding of the IVI to overcome this challenge (Fig. 1f). The overall approach involves integrating IVI with surgical catheters by virtue of the elastic deformability of the silk scaffold and its compatibility with dMEA in the miniaturization step, enabling targeted implantation and concurrent recording of deep brain nuclei while draining CSF externally. This flexible dMEA fabrication technique, combined with a silk-enabled minimally invasive implantation strategy, can produce a silk-enabled IVI compatible with commonly used neurosurgical catheter dimensions (2.2 mm inner diameter) (Supplementary Fig. 3). Furthermore, owing to the in-situ drug delivery property of silk fibroin, drugs required for electrode implantation and neurosurgical procedures, such as dexamethasone, can be introduced to alleviate acute neuroinflammatory reactions for IVI. After fabricating the silk-enabled self-unfolding IVI and completing catheter assembly, biocompatibility was prioritized and assessed. Specifically, all components in contact with neural tissue—including the dMEA, silk scaffold, tantalum marker, and soldering pads of the FPC—were implanted epidurally in mice. Immunohistochemical analysis, as shown in Supplementary Fig. 4, confirmed the IVI's good biocompatibility.

### Silk-enabled minimally invasive implantation and self-unfolding strategy

The primary reason for incorporating silk scaffolds is to enhance the IVI's conformal attachment capability to deep brain nuclei. This enhancement primarily stems from the self-unfolding ability of the IVI, attributed to the shape memory characteristic of the silk scaffold[34–36] (Fig. 2a, see Supplementary Note 1 for the detailed mechanism). During the miniaturization of the IVI, the top of the silk scaffold was compressed while the bottom was tensioned. This direction-oriented crystallization can be observed in 2D wide-angle X-ray diffraction (2D-WAXD) patterns, which clearly show the evolution of the diffraction ring to diffraction arcs (Fig. 2b). The 2D-WAXD diffraction arcs of the compression sample are perpendicular to the bending direction, while those of the tension sample are parallel to the bending direction. This glassy, temporarily fixed state can revert to the original elastomeric state after CSF-triggered self-unfolding, which was also observed in the 2D-WAXD patterns of re-dried samples (Supplementary Fig. 5). When the IVI is being implanted in the deep brain target area, the hydrogen

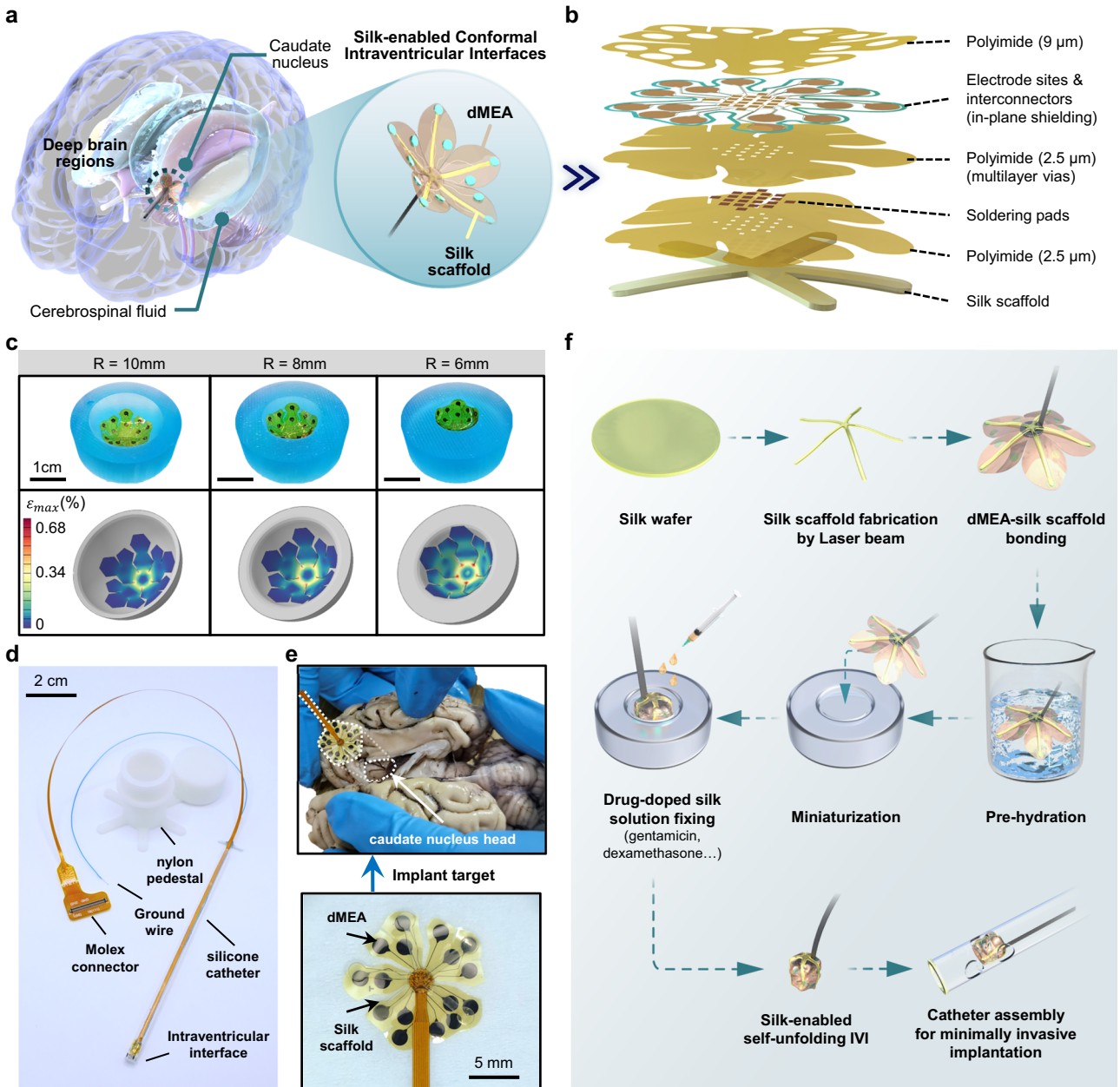

**Fig. 1 | Overview of the silk-enabled conformal intraventricular interface (IVI).**
**a** Schematic of the silk-enabled minimally invasive IVI for interfacing subcortical nuclei, such as the caudate nucleus, in the deep brain regions, showing IVI conformally attaches to the surface of the caudate head in a self-unfolded form when being immersed in the CSF. The zoom-in figure shows the two main functional parts of the IVI. **b** An explosive view of IVI configuration. The electrode sites and soldering pads are located on both sides of the device through multilayer vias, the metal shielding layer is coplanar with the layer of electrode sites and interconnectors, and the silk scaffold is attached to the bottom polyimide. **c** Mechanic

finite element analysis of the IVI microelectrodes conformally attached to different curved surfaces, the upper panel showing the corresponding photos of IVI attached to the same curved surface mold. **d** The photo of the complete IVI system, including the implant and the skull-mounted base. **e** The image of the unfolded state of the implant device, including the small-sized dMEA and the corresponding silk scaffold. The upper image shows the caudate nucleus head being pointed as the implantation target. **f** Flow chart showing the preparation and assembly of the catheter-integrated minimally invasive IVI from silk wafer and dMEA.

bonds are broken down by the CSF, allowing the top and the bottom parts of the silk scaffold to revert to their original elastomeric states in the opposite direction, which demonstrates the self-unfolding effect of the IVI.

Based on this self-unfolding mechanism of silk scaffolds and the need for IVI to conformally attach to neural nuclei surfaces with different curvatures, we designed two types of IVI models—convex and concave. The distinction mainly lies in the direction of bending of the silk fibroin scaffold during the miniaturization step and its fixation method with the dMEA, allowing for conformal attachment to convex

and concave surfaces in the cerebrospinal fluid environment, as validated in our custom-made 1:1 scale 3D-printed transparent lateral ventricle model (see Fig. 2c and methods). The minimally invasive implantation and conformal attachment in vitro validations demonstrated that the folded IVI can spontaneously self-unfold upon protruding from the catheter into the lateral ventricles (Supplementary Movie 1). To assess the impact of potential micromovements on the stability of the IVI–neural tissue interface, we simulated deep brain nuclei using a 0.6% agarose model with curved surfaces of 7 mm and 10 mm in diameter. Results showed that after self-unfolding and

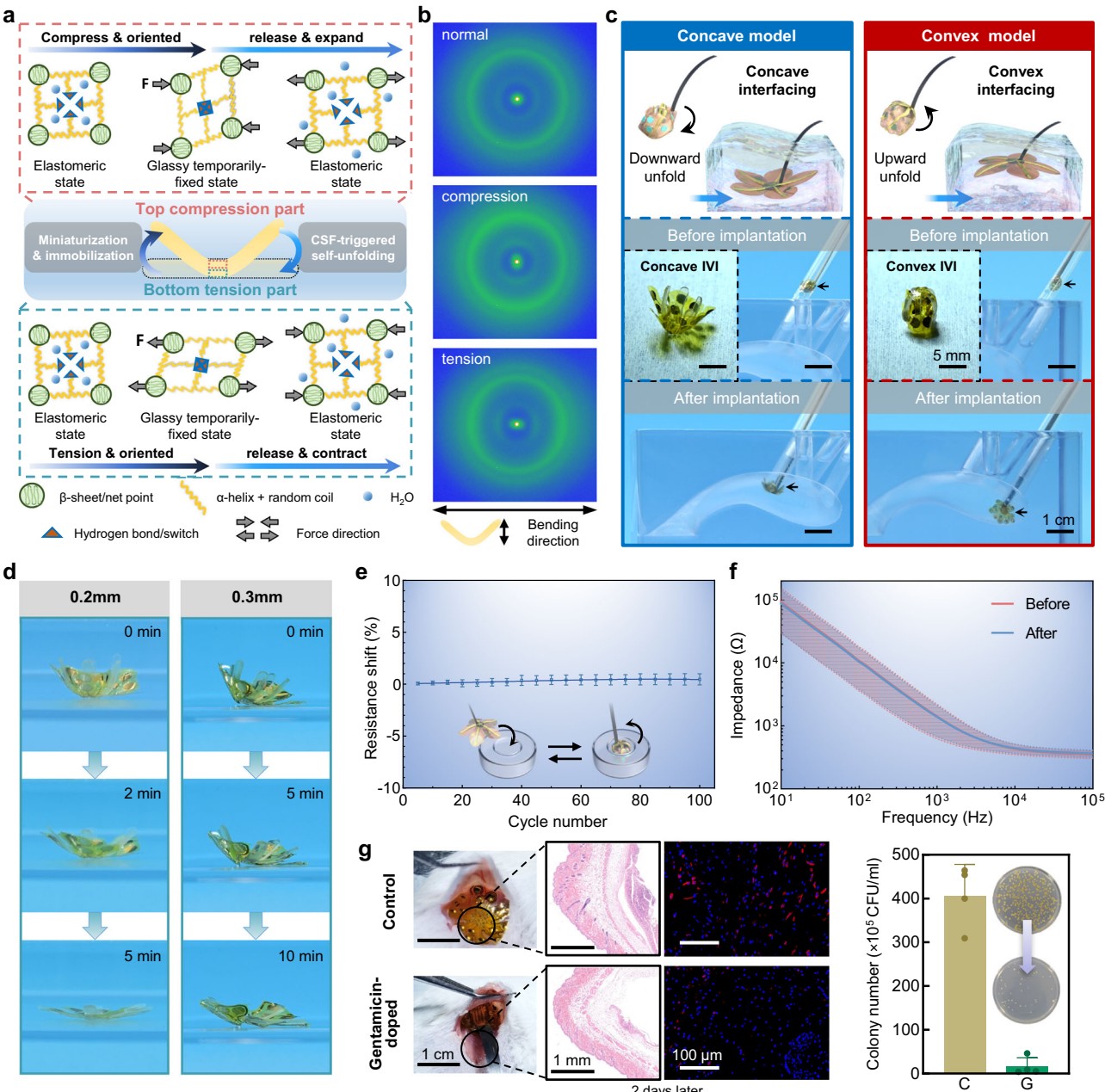

**Fig. 2 | Silk-enabled minimally invasive implantation and self-unfolding strategy of IVI. a** Schematic view of the micro mechanism of the miniaturization and CSF-triggered self-unfolding of the silk scaffold. **b** 2D-WAXD patterns of the normal, compression and tension states of the silk scaffold. **c** Images and photos showing the application design and in vitro conformal interfacing and implantation validation of the convex and the concave models of IVI. Insets show the different configurations of these two IVI models. **d** Photos of IVIs with 0.2-mm and 0.3-mm-thick silk scaffold achieved self-unfolding within 5 min and 10 min, respectively, in artificial CSF at 37 °C. **e** The direct current resistance fluctuations of IVI microelectrodes arising from 100 cycles of bending and warping. ($n = 48$ electrode sites from 3 devices). **f** Impedance curves of the IVI microelectrodes before and after the validation in (**c**) ($n = 48$ electrode sites from 3 devices). **g** In vivo experiments of drugs-loaded IVI for drug release in mice, with photographs of implants in mice back of the control group and the gentamicin drug release group, HE tissues staining images, immunohistochemistry images, as well as in vitro solid culture photographs of tissue fluid from the wounds after 2 days and a comparison of colony counts ($n = 4$ from 4 mice, $16 \pm 10.06$ versus $406.8 \pm 35.49$). Data are presented as mean values ± SD in (**e**) and (**f**), and as mean + SD in (**g**). Source data are provided as a Source Data file.

conformally attaching to the 7 mm surface with silk scaffold assistance, the IVI could further self-unfold and conform to the 10 mm curvature (Supplementary Fig. 6). These findings demonstrate the IVI's dynamic adaptability for stable conformal attachment within the pulsatile cerebrospinal fluid environment, facilitated by the silk scaffold's shape recovery process and its interaction with tissue surfaces. Different implantation routes determined by surgical requirements have also been validated on the transparent lateral ventricles model

(Supplementary Fig. 7). During the phase of silk scaffold-assisted self-unfolding, the speed of IVI self-unfolding can be controlled by adjusting the thickness of the silk scaffold, thereby accommodating different time windows for various surgical protocols (Fig. 2d and Supplementary Movie 2).

Maintaining stable electrical performance and a reliable IVI-neural interface is crucial for IVI operation. Direct current resistance tests under cyclic stress showed resistance drift remained within 5% after 100 cycles

(Fig. 2e), while impedance spectra before and after assembly confirmed no electrical degradation (Fig. 2f). To assess long-term stability for potential prolonged use, impedance tests over 4 weeks in artificial CSF at 37 °C demonstrated robust encapsulation between the dMEA and FPC (Supplementary Fig. 8). In minimally invasive neural interface implantation, in situ drug delivery enhances specificity, localizes drug action, and minimizes systemic side effects[37,38]. To assess the efficacy of drug release from the drug-loaded IVI, we conducted subcutaneous antibacterial experiments in mice (Fig. 2g). Both the gentamicin-treated and control groups were infected with equal doses of *Staphylococcus aureus*. After 2 days, immunohistochemistry showed reduced macrophages in the gentamicin group, while agar culture confirmed significantly fewer bacterial colonies.

### In-plane shielding for high-quality electrophysiological signal recording

Effective electrostatic shielding is also essential for acquiring high-quality neurophysiological signals[28], especially for the neural interface in deep brain regions (see Supplementary Note 2 for details about shielding). Hence, we proposed the in-plane metal shielding design within the existing conductor layers, without altering the device's hierarchical structure or compromising the compliance of neural electrodes. The COMSOL software was employed to construct a simplified model and conduct electrostatic field simulations. The simulation results for normalized potential, as illustrated in the contour maps, demonstrate the efficacy of coplanar metal shielding (Fig. 3a). We infer that the T1-type IVI microelectrodes, featuring an inner clock-like and an outer irregular annular shielding design, will exhibit a significant reduction in susceptibility to external electromagnetic interference, in comparison to the unshielded T0-type IVI design (Fig. 3b and Supplementary Fig. 9a). Numerical analysis of the normalized potential at the dashed line in cloud maps revealed a minimum 50% reduction in potential at microelectrode sites and interconnectors (Fig. 3c).

Two configurations of IVI (T0 and T1) were immersed in phosphate-buffered saline (PBS) to test the baseline noise. Noise density spectrum indicated that the T1-type IVI exhibited lower noise levels in the frequency range below 10 Hz and at power frequency and its harmonics compared to the T0-type IVI (Fig. 3d). Statistical comparisons of noise density at the power fundamental frequency and all harmonics within the 200 Hz range revealed that the T1-type IVI made a significant improvement over the T0-type IVI (Fig. 3e). Statistical analysis of the root mean square (RMS) noise of the original signals further confirmed that the T1-type IVI reached optimal system noise levels when properly grounded (Supplementary Fig. 9b and Supplementary Fig. 10). In vivo electrophysiological experiments were further conducted. Subdural cortical recordings by T0- and T1-types of IVI microelectrodes were performed on the penicillin-induced epileptic mouse model via intraperitoneal injection. Typical epileptiform discharges during the seizure onset were observed (Fig. 3f), and time-frequency analysis of the raw signals (Fig. 3g) revealed that the T1 spectrogram exhibited reduced 50 Hz power frequency interference and more prominent epileptiform time-frequency characteristics. This feature helps prevent researchers from misinterpreting noise introduced by mechanical artifacts as epileptiform discharges during initial manual screening of raw data. The power spectral density (PSD) analysis results also indicated that compared with T0, T1 exhibited lower energy at the power frequency and its harmonics. The lower energy in the low-frequency signals was consistent with in vitro validation, suggesting that it was less susceptible to interference from potential mechanical artifacts (Fig. 3h). The computed signal-to-noise ratios (SNR) of the signals acquired from the epileptic mouse by two types of IVIs showed that T1 presented a higher SNR than T0 [6.626 ± 0.1119 versus 4.217 ± 0.1395] (Fig. 3i). Furthermore, we performed subdural cortical recordings in anesthetized Labrador dogs and conducted a comparative analysis of the quality of raw signals collected by T0 and

T1. The results consistently demonstrated that T1 exhibited significantly lower power frequency noise and system baseline noise (Supplementary Fig. 11).

### Interfacing and decoding the neural activities from the Parkinsonian sheep

We further explored the application of IVI in a chronic Parkinsonian sheep model for preclinical validation, endeavoring to establish a flexible neural interface on the surface of the head of the caudate nucleus (see Supplementary Note 3, Supplementary Figs. 12–15 for more details about the preclinical model). Intraoperative computed tomography (CT) scan images of the sheep brain, both in coronal and horizontal planes, displayed the CT markers on the IVI located near the caudate nucleus head (Fig. 4a and Supplementary Fig. 16; see methods for details about the surgery), thereby confirming the success of the targeted implantation. The recorded original electrophysiological signals showed minimal mechanical artifacts and power frequency interference (Fig. 4b), preliminarily verifying the effectiveness of the IVI. Representative immunohistochemical staining showing a reduction of dopaminergic neurons in the substantia nigra following MPTP-induced Parkinsonian model establishment in sheep (Fig. 4c). Comparable staining patterns were observed in the other two animals examined, indicating consistency across all three cases (see Methods for details about model establishment).

After a period of effective recording from the head of the caudate nucleus using the IVI, Levodopa (LD) combined with Benserazide was injected subcutaneously as planned (Supplementary Fig. 17). This pharmacological combination has been shown to induce the reversal of parkinsonism in previous studies[39]. Following the subcutaneous injection, another period of effective recording was maintained. Offline analyses related to the pathological neural activity of the basal ganglia in PD showed an increase in beta-band energy during the Before LD phase compared to the After LD phase, and a notable beta energy peak around 19 Hz was observed in the Before LD phase ($n = 4$) (Fig. 4d). Supplementary Fig. 18 showed the beta burst detection method, and the duration and amplitude of all detected beta bursts during the Before LD and After LD phases are illustrated in Fig. 4e. It showed a strong correlation in both phases between the duration and amplitude of bursts [Before LD: Spearman's rho (rs) = 0.9395; After LD: rs = 0.9062], which was evident across several channels that exhibited significant beta burst activities (Supplementary Fig. 19). The duration of beta bursts was then divided into seven windows (bins) and the proportion of beta bursts within each duration window relative to all beta bursts was calculated. Compared to the After LD phase, there were fewer short-duration beta bursts (0.1–0.15 s) in the Before LD phase. Conversely, long-duration beta bursts (0.3–0.4 s; 0.4–0.5 s; >0.5 s) were more prevalent in the Before LD phase than in the After LD phase (Fig. 4f). Further, statistical analyses were conducted on several characteristics of beta bursts during both phases, including the average burst amplitude, average duration, and burst probability. The results revealed that all three characteristics of beta bursts were higher during the Before LD phase (Fig. 4g). These characteristics of beta burst activities were observed across more than one channel (Supplementary Fig. 20 and Supplementary Fig. 21), showing that in the MPTP-induced Parkinsonian sheep model, neural activity at the caudate nucleus surface is also dominated by long-duration beta bursts. These findings indicate that abnormal oscillatory activity associated with the Parkinsonian state, as well as the electrophysiological effects of dopaminergic replacement therapy, were successfully captured by the IVI at the surface of the caudate nucleus.

A singular value decomposition and linear discriminant analysis (SVD-LDA) model was established to discriminate different Parkinsonian pathological neural activity states (see methods and Supplementary Fig. 22a–c). The neural activity discrimination accuracy across all effective channels was computed. Channels 5 and 9 exhibited higher accuracies, 90.3% and 89.9% respectively (Fig. 4h). Additionally,

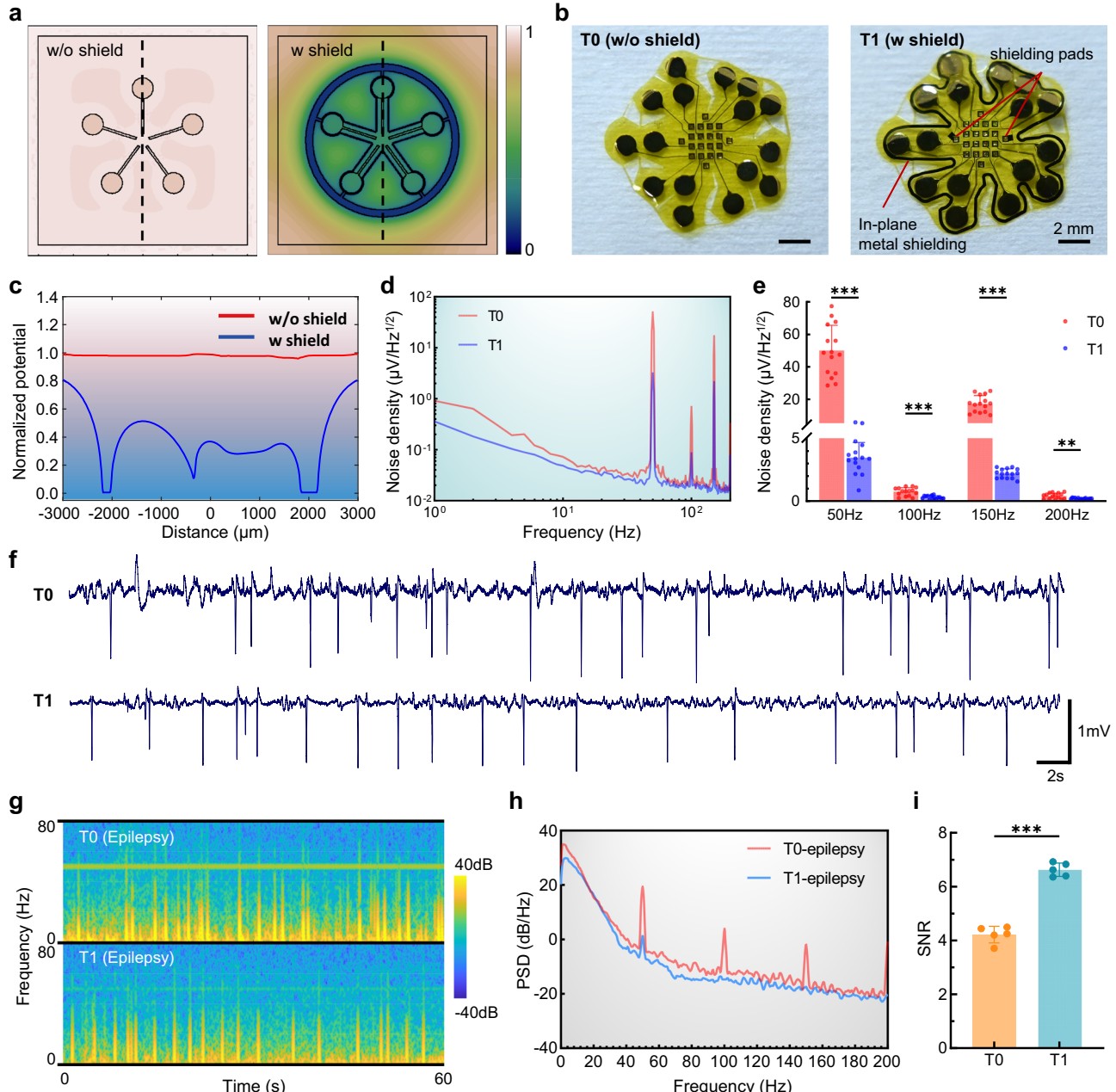

**Fig. 3 | High-quality electrophysiological recording of IVI with in-plane shielding. a** Heat maps of electrostatic field simulation showing the normalized potential distribution in a simplified microelectrode model with and without an in-plane metal shielding layer. **b** Images of the T0-type (left, without shield) and T1-type IVI microelectrodes (right, with shield) corresponding to the simulation design in (**a**). **c** Normalized potential distribution at the cross-section in (**a**). **d** Noise density spectrum of T0- and T1-types of IVI microelectrodes. **e** Bar plots showed the noise density at the power frequency and its harmonic frequencies of two types of IVI microelectrodes. $n = 15$ microelectrodes. Multiple unpaired two-tailed $t$ tests: 50 Hz, $p = 0.0000$; 100 Hz, $p = 0.0002$; 150 Hz, $p = 0.0000$; 200 Hz, $p = 0.0043$.

**f** Representative epileptiform discharge waveforms acquired during the seizure onset of a penicillin-induced epilepsy mouse model, recorded by T0- and T1-types of IVI. **g** Typical electrophysiological spectrograms of 0-80 Hz, revealing the time-frequency characteristics recorded by T0 and T1 during 60 s epileptic seizures. **h** Power spectral density plots illustrating reduced low-frequency mechanical artifacts and power line interferences of T1's signal compared to T0's signal. **i** Scatter dot plots showing the signal-to-noise ratio of T0 and T1 from the epilepsy recordings ($n = 5$ sessions of recordings). Unpaired two-tailed t tests: $p = 0.0000$. Data are presented as mean values + SD in (**e**), and as mean ± SD in (**i**). Source data are provided as a Source Data file.

combined data from channels 5 and 9 improved the average accuracy to 91% (Supplementary Fig. 22d). Considering a random discrimination accuracy of 33.3%, a significance distribution of discrimination accuracies across channels on the IVI was also obtained (Supplementary Fig. 23a). To confirm the independence of channels in neural discrimination, the Pearson correlation coefficients among signals of IVI channels were calculated (Fig. 4i and Supplementary Fig. 23b). Notably, the Pearson correlation coefficients between channels 5 and 9

in the chronic parkinsonian state indicate that these two channels with high discrimination accuracies largely provide uncorrelated neural signals, implying the potential for further extraction of different underlying neural features.

## Intraventricular electrophysiology in freely moving sheep

Intraoperative recordings from the caudate nucleus surface in Parkinsonian sheep validated the feasibility of minimally invasive

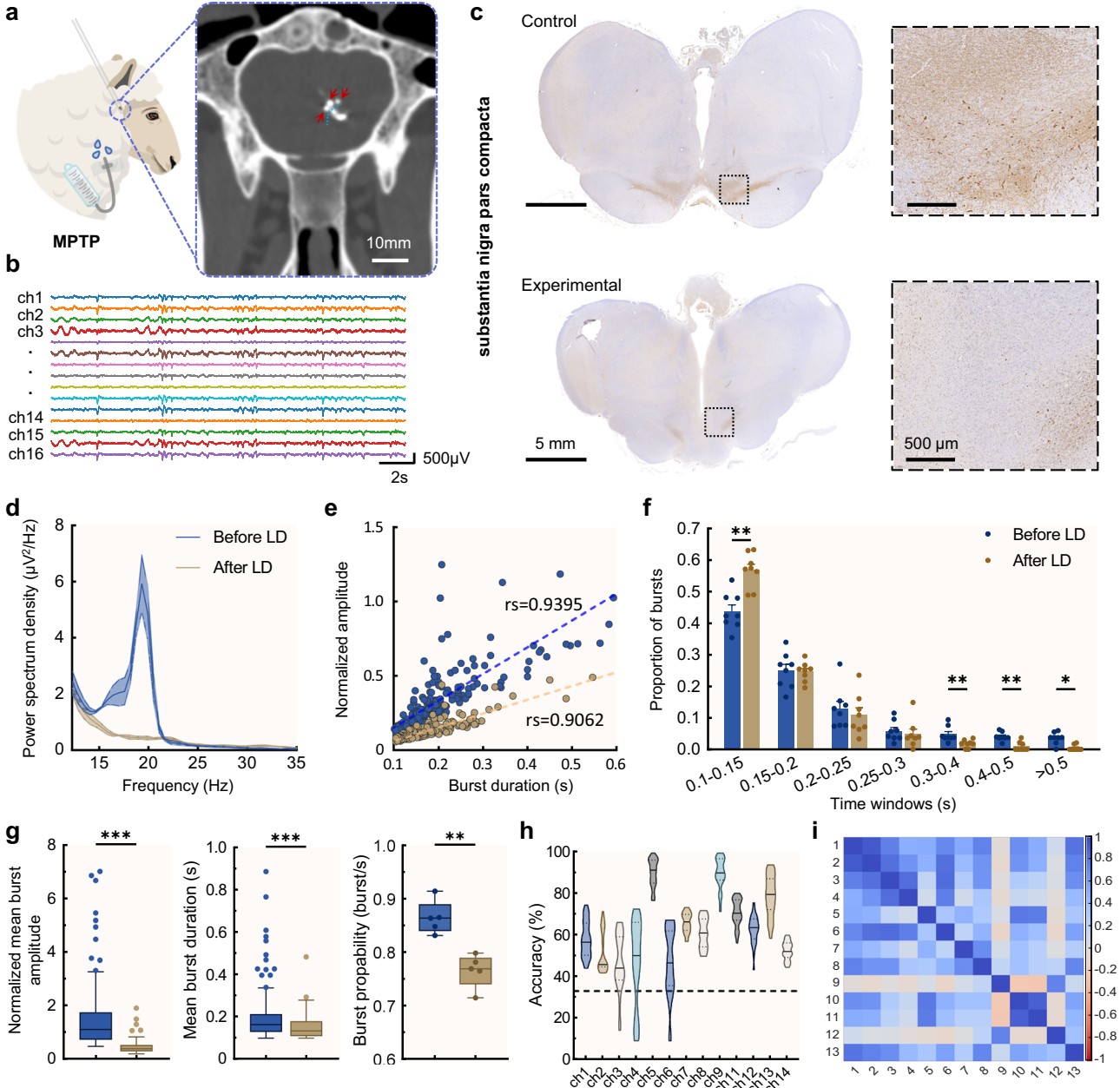

**Fig. 4 | Intraventricular electrophysiological recording and neural decoding in Parkinsonian sheep. a** Illustration of MPTP-induced parkinsonian sheep via jugular infusion, and intraoperative CT image of IVI implantation. Red arrows indicate markers on the recording sites, and the blue dashed line indicates the contour of the caudate head. **b** Typical neural activity recorded from the caudate nucleus surface. **c** Representative immunohistochemical staining showing a reduction of dopaminergic neurons in the substantia nigra following parkinsonian sheep model establishments. **d** Power spectral density of local field potentials recordings (mean ± SD) (*n* = 4 recordings), with a beta peak at 19 Hz in the Before LD phase and a reduction of beta power in the After LD phase. **e** Amplitude and duration of all detected beta bursts of both phases from five minutes recordings, respectively. **f** Scatter dot plots showing changes in distribution of burst durations as a percentage of total number of bursts (mean + SEM), during Before LD and After LD phases. Two-sided multiple Mann–Whitney tests: *n*1 = *n*2 = 8 sessions of recordings,

0.1–0.15 s, *p* = 0.0043; 0.3–0.4 s, *p* = 0.0056; 0.4–0.5 s, *p* = 0.0056; >0.5 s, *p* = 0.0130. **g** The normalized mean burst amplitude during Before LD and After LD, *t*(283) = 9.792, *p* = 0.0000. The mean duration of all bursts during Before LD and After LD, *t*(283) = 3.824, *p* = 0.0002. *n*1 = 147, *n*2 = 138 detected beta bursts. The probability of bursts to occur (illustrated as burst/s) was reduced after levodopa administration (*n* = 5 sessions of recordings), *t*(8) = 5.034, *p* = 0.0010. Two-sided unpaired *t* tests were used. Same color code as (**f**). **h** The discrimination accuracy of all effective IVI channels for the three states, chronic Parkinsonian, pathological beta burst, and post-levodopa administration. **i** Topological maps of Pearson correlation coefficients for ten minutes of raw data in the chronic Parkinsonian state from all effective IVI channels. The box plots and truncated violin plots show the median and interquartile range, and the whiskers denote 1.5× the interquartile range. Source data are provided as a Source Data file.

implantation of the IVI and demonstrated its stable functionality within the CSF of the lateral ventricle. To address clinical demands for postoperative monitoring and neuroscientific requirements for long-term tracking of neural circuit dynamics, chronic in vivo experiments were conducted using the small-sized IVI (see Methods and Supplementary

Fig. 3) in the chronic Parkinsonian sheep model to further evaluate its translational potential.

The experiment was designed to track dynamic changes in periventricular electrophysiology before, during, and after the establishment of an MPTP-induced PD model, over a timespan comparable to

the clinical implantation period of SEEG electrodes (Supplementary Fig. 24). A surgical procedure adapted from the intraoperative recording experiments was employed for IVI implantation in the chronic in vivo studies, with critical modifications involving the sealing and fixation of the IVI's backend and anchoring of the skull-mounted base with bone screws (see details in Methods). Following surgery, an Elizabethan collar was applied to provide additional protection for the implant and surgical site (Fig. 5a), and electrophysiological recordings were performed at scheduled intervals while the sheep remained freely mobile (Fig. 5b).

Long-term functionality, mechanical stability, biocompatibility, and electrical robustness of the miniaturized IVI were assessed afterwards to support the evaluation of its translational potential. Weekly recorded signals were first aggregated and analyzed using power spectral density (PSD) methods, revealing a prominent -18 Hz beta-band peak following the establishment of chronic PD (Fig. 5c). The dataset was subsequently segmented into three experimental stages—postoperative, during modeling, and chronic PD—for beta burst analysis using a comparable approach (Supplementary Fig. 18). Scatter plots illustrating the amplitude and duration of all detected beta bursts across the three stages during valid recording sessions are presented in Fig. 5d. Notably, the beta burst characteristics observed in the chronic PD state closely resembled those identified in the intraoperative recordings, exhibiting increased amplitudes and prolonged durations. Distinct differences in beta burst characteristics before and after PD induction, as well as their consistent evolution throughout chronic modeling, were revealed by windowed distribution analysis of burst durations (Fig. 5e). Statistical evaluation further confirmed a progressive increase in burst amplitude, duration, and frequency, reflecting the gradual development of the chronic PD model (Fig. 5f–h).

Mechanical stability was confirmed by comparing intraoperative and postoperative CT scans, which showed no detectable displacement of the IVI relative to the lateral ventricle and a gradual restoration of ventricular morphology (Fig. 5i). For long-term biocompatibility, terminal immunofluorescence analysis was performed at the endpoint of four-week implantation studies. Quantitative comparisons of NeuN-labeled neurons, Iba1-labeled microglia, and GFAP-labeled astrocytes between the IVI-implanted and contralateral hemispheres in coronal sections demonstrated the absence of significant immune rejection after four weeks of implantation (Fig. 5j). Finally, a statistical evaluation of impedance and signal-to-noise ratio (SNR) across the four-week continuous recording period confirmed the long-term stability and reliability of the IVI's electrical performance (Supplementary Fig. 25). These findings establish the IVI as a reliable neural interface for chronic recordings from the surfaces of periventricular neural structures, enabling the detection of state-dependent neural dynamics and their modulation in response to external stimuli.

## Discussion

Current flexible planar microelectrode arrays, despite their large coverage and conformal attachment, cannot access the brain's inner surface, and establishing a stable neural interface within the dynamic CSF environment remains a major challenge for intracranial electrodes. Our design theoretically enables large-scale neural recordings from all CSF-facing surfaces of deep brain structures, while maintaining a safety and duration profile comparable to clinical intracranial electrophysiology, such as electrocorticogram and SEEG. By integrating a flexible, self-unfolding neural interface into neurosurgical catheters used for ventricular access, this approach achieves conformal attachment to the ependyma and facilitates minimally invasive neural recordings from periventricular structures surfaces within the dynamic CSF milieu. Intraoperative and chronic experimental validations in Parkinsonian sheep further demonstrated the feasibility, effectiveness and translational utility of this technology.

This monitoring strategy provides a complementary solution to existing macro- and microelectrode technologies, such as SEEG and brush-style electrodes, which would cause invasive damage to the targeted neural nuclei[15,20,40]. While these conventional electrodes are effective for volumetric recordings, they are less suitable for stable, large-area recordings from inner surfaces of the brain within the CSF environment. In contrast, our approach prioritizes minimally invasive access through a conformal, structurally adaptive interface specifically designed for electrophysiological monitoring of periventricular neural structures. Although recent advances in endocisternal interfaces provide alternatives with reduced invasiveness, their functionality remains largely confined to co-axial neural recordings[41].

Future iterations of this technology will aim to achieve seamless integration with standard tools employed in minimally invasive neurosurgery, including endoscopes and external ventricular drainage catheters. In parallel, coupling the IVI's backend with that of cortical electrodes may enable simultaneous recordings of intraventricular and cortical neural activity. We envision this platform as a valuable clinical tool, enabling researchers to chronically monitor periventricular neural circuits and to identify biologically relevant stimulation targets and parameters. Such advances could open new avenues for real-time monitoring, early warning, and therapeutic intervention in neurological and psychiatric disorders involving periventricular brain structures. Furthermore, by establishing a stable long-term neural interface on the brain's inner surface, it has the potential to advance the study of information exchange and functional connectivity between surface neurons in deep brain regions and distant brain areas.

## Methods

### Fabrication of deformable microelectrode array

As the core component of IVI, the 14 μm-thick polyimide-based dMEA on Si substrates was fabricated using a microelectromechanical systems (MEMS) process that builds on our previously reported dual-sided interconnection platform—combining double-sided metal exposure with interlayer vias to link a back-side reflow-pad array to front-side recording sites[42,43]. For the present IVI, we adapted the layout to a hemispherical-unfolding pattern with rounded corners (circular sites, 700 μm radius) and implemented two variants: T1 with a coplanar Cr/Au shield on the interlayer polyimide (PI) and T0 without shielding. Regarding catheter scalability, a conventional version was implemented for preliminary experiments, with the dMEA center measuring 2.32 × 2.28 mm and a 0.6-mm pad pitch. To further enhance clinical viability and reduce the footprint, a small-sized IVI was developed using flip-chip bonding (Finetech GmbH & Co. KG; 160-μm pad pitch) and multilayer FPC optimization, yielding miniaturized components (dMEA center: 1.27 × 1.27 mm) for the long-term in vivo study. The microfabrication flow remained identical to that described below. Aluminum (Al) (1 μm) sacrificial sputter; PI base (3 μm) patterning and cure; back-side pad metallization by e-beam evaporation (Cr/Ni/Au: 100/1000/5000 Å) and liftoff; PI interlayer (2 μm) with via openings; via/shield metallization (Cr/Au: 150/4000 Å; shield omitted for T0); front-side traces and recording sites (Cr/Au: 150/4000 Å); PI encapsulation (8 μm) with matched site windows and mesh-style strain-relief features; Al release in buffered hydrofluoric acid (see Supplementary Fig. 1). Compared with sandwich-type flexible stacks with extended on-chip leads[44], this architecture enables a smaller footprint, higher site yield, and reliable pad-to-site connectivity conducive to minimally invasive assembly[43].

### Simulation design of IVI

Commercial mechanical simulation software ABAQUS 2020 was utilized to perform mechanical simulations on the attachment of IVI microelectrodes to surfaces with varying curvatures. The microelectrodes were modeled using shell elements, with S4R quadrilateral shell elements used for mesh division. Static analysis was conducted using

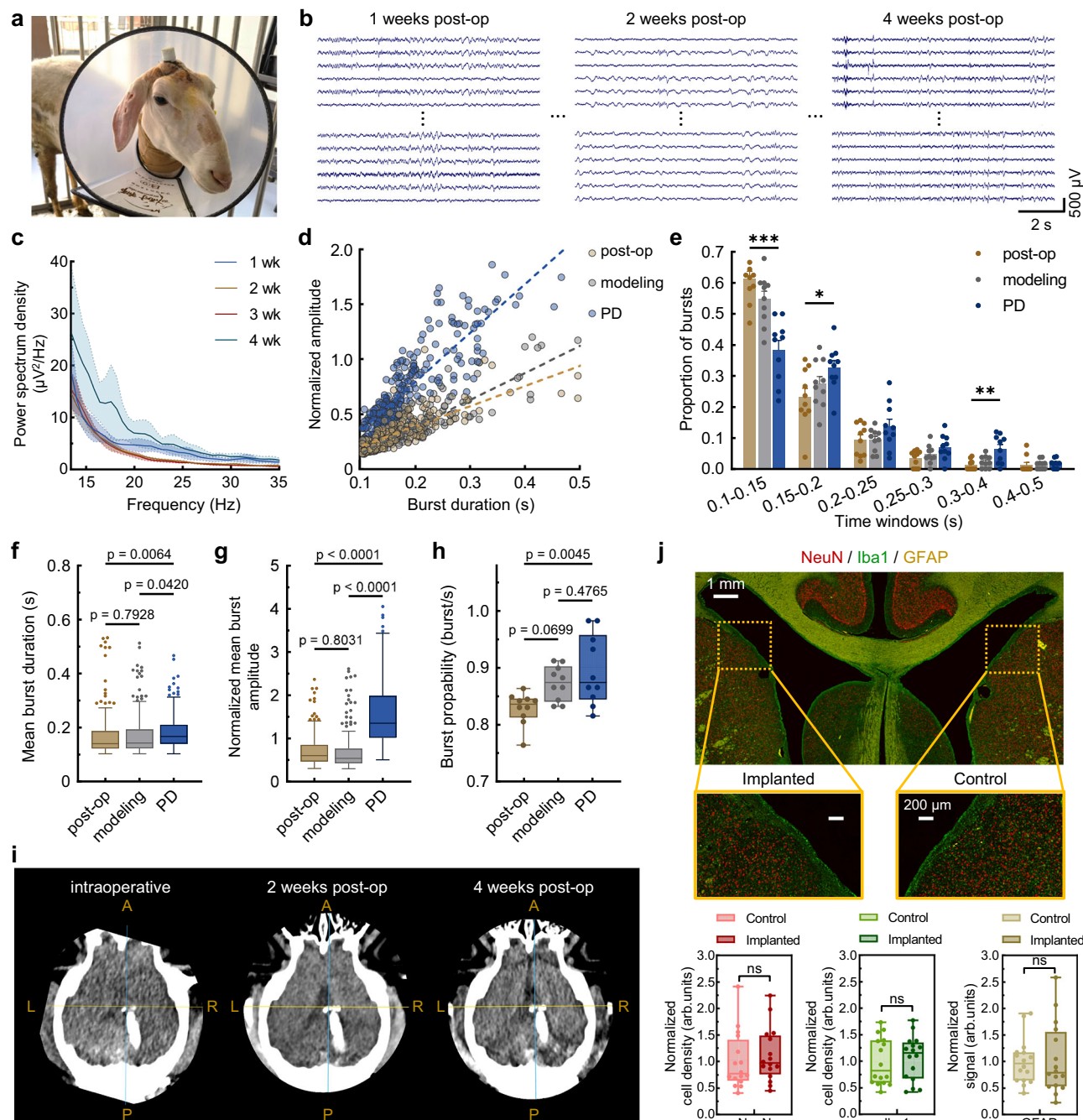

**Fig. 5 | Intraventricular electrophysiology in freely moving sheep. a** The photo of the free-moving sheep after surgery. During post-operative care and recovery, the implant and surgical incision are further protected with a pedestal and an Elizabethan collar. **b** The waveforms of neural activity recorded by the IVI on free-moving sheep during four weeks. **c** Power spectral density of local field potentials recordings (mean ± SD) (*n* = 5 recordings), with a beta peak at 18 Hz during recordings at the fourth week. **d** Beta burst amplitude and duration were extracted from 5-min artifact-free recordings obtained during each of the three phases: post-operative, MPTP modeling, and the Parkinsonian state. **e** Scatter dot plots showing changes in distribution of burst durations as a percentage of total number of bursts (mean + SEM) (*n* = 10 recordings), during these three phases. Two-way ANOVA tests (two-sided), followed by Dunnett's post hoc tests: 0.1–0.15 s, *p* = 0.0000; 0.15–0.2 s, *p* = 0.0360; 0.3–0.4 s, *p* = 0.0088. **f** The duration of all bursts during three phases, $F(2, 759) = 5.221$, $p = 0.0056$. **g** The normalized mean burst amplitude

of all bursts, $F(2, 759) = 227.5$, $p = 0.0000$. **h** The probability of bursts to occur (illustrated as burst/s), $F(2, 27) = 6.335$, $p = 0.0055$. Values are represented as mean + SEM. Two-sided one-way ANOVA tests of beta bursts characteristics (*n* = 254 detected bursts for (**f**, **g**); *n* = 10 recordings for (**h**)). **i** Intraoperative and follow-up CT scans demonstrated the restoration of lateral ventricle morphology and the mechanical stability of the IVI, with no apparent displacement.
**j** Immunofluorescence images showing the periventricular structures on the IVI-implanted and contralateral control sides of the sheep brain. Quantitative analysis included the normalized cell density of NeuN-labeled neurons (*p* = 0.2157), the normalized cell density of Iba1-labeled microglial cells (*p* = 0.6497) and the normalized signal of GFAP-labeled astrocytes (*p* = 0.8285). Two-tailed paired t tests were used. *n* = 16 immunofluorescence sections. The box plots show the median and interquartile range, and the whiskers denote 1.5× the interquartile range. Source data are provided as a Source Data file.

the Standard module, and the microelectrode was set to hard contact with the curved surface. Finite element analysis was completed by setting displacement boundary conditions to attach the center of the microelectrode to a specified surface. This simulation aimed to model the real strain distribution of the microelectrode when conforming to neural nuclei with specific curvatures.

To guide the coplanar metal shielding design for the deep-brain neural interface, we employed commercial multiphysics simulation software COMSOL 6.0 to conduct electrostatic field simulations on a simplified model of the IVI microelectrode. The entire model consisted of a metal layer encapsulated by two layers of PI, with the metal shielding surrounding the microelectrode leads on the same plane. The simulation was set in a PBS solution environment, with the potential at the top layer of the PBS solution model set to specific potential values, and the bottom layer potential set to 0. This setup aimed to simulate interference from active devices that could occur at any location above the recording microelectrode sites.

### Fabrication of silk scaffolds
Initially, pre-weighed lyophilized silk fibroin powder was finely chopped and placed in a polyethylene syringe, then dissolved in a specific volume of HFIP solution (Sigma-Aldrich). The volume of the solution was calculated based on the desired final film thickness and the density of the thick silk film ($1.3\,g/cm^3$). The silk fibroin solution, after several hours of dissolution, was slowly poured into a flat-bottomed polyethylene container, creating a sub-seal condition to facilitate smooth evaporation. After several days of slow evaporation, the partially formed thick silk fibroin film was removed and left to dry and flatten in a splint. The fully formed silk fibroin film was then sequentially immersed in a methanol solution and pure water for several days. The silk fibroin wafer, now devoid of organic solvents, was dried and flattened in a splint again. The patterning of the silk wafer was accomplished through laser processing (ProtoLaser U3, LPKF Laser & Electronics, Germany), and the silk scaffolds obtained were immersed in pure water for subsequent assembly use (Supplementary Fig. 2).

### Assembly of silk-enabled minimally invasive IVI
The reflow soldering pads of microelectrodes were encapsulated with a biocompatible ultraviolet (UV) curable adhesive (MD®1128A-M-VT, DYMAX, USA). The IVI device was obtained by bonding and fixing the center-aligned silk scaffold, which had an initial planar morphology, with the prepared IVI microelectrodes using the same adhesive. Subsequently, IVI was immersed in pure water for 30 min for pre-hydration. Next, the IVI was carefully placed into home-made PDMS molds for miniaturization and left to air dry at room temperature (25 °C) for 12 h or in a convection oven at 30 °C for 60 min to fix its shape. It is noteworthy that the morphology at this step represents the folded state of IVI in the surgical catheter, determining the dynamic shape change process upon recovery to its initial state. To this end, we designed two models of IVI (Supplementary Fig. 7, Supplementary Movie 3, and Supplementary Movie 4), and their miniaturization was achieved either through PDMS or slightly undersized catheters. The drug-doped silk fibroin solution was pipetted onto the miniaturized IVI in PDMS molds, with careful control to apply the solution onto the silk scaffold, followed by incubation at room temperature for 2 h. The drug-loaded IVI were carefully removed from molds and assembled into silicone surgical catheters with an inner diameter of 5 mm, for large animal validations. To enable further clinical applications with minimal implantation-induced damage, a compact dMEA and its corresponding FPC were designed and fabricated. The resulting small-sized IVI is compatible with surgical catheters featuring an internal diameter of 2.2 mm (Supplementary Fig. 3). The degradable silk membrane sealing at the surgical catheter tip was intended to minimize the influence of cortical tissue on IVI during the minimally invasive implantation process. The small openings on the catheter's end

sidewall are standard design features to balance intracranial and extracranial pressure differentials and provide a pathway for CSF drainage. Additionally, some sites of the IVI (such as the end of the silk scaffold and the end sidewall of the surgical catheter) were marked with tantalum powder fixed by dotted UV adhesive for enhancing visibility and imaging under CT as required by the surgical protocol.

### Optical characterization of silk scaffolds
2D wide-angle X-ray diffraction (2D-WAXD) was performed with a Xeuss 3.0 instrument (Xenocs, France) with an X-ray of 0.154 nm wavelength (50 kV, 0.6 mA) in the range of 2 theta of 3.5–34. The normal sample was not subjected to the miniaturization process. The compression and tension part samples were cut from the center of the silk scaffold, which has the greatest bending stress after the miniaturization. The re-dried samples refer to those that underwent water-triggered self-unfolding and were subsequently dried freely without any restriction, and the normal sample underwent the same water immersion process.

### Animals
Adult male C57BL/6 mice (10 weeks old) were used for the in vivo in-plane shielding validation experiments. Mice were provided by the Shanghai Laboratory Animal Research Center (SLARC). Two or three mice were housed in a cage that had a 12:12 light–dark cycle. The temperature and humidity of the animal facility were maintained at 23 ± 2 °C and 50 ± 5%. Ethical approval for our animal experiments was obtained from the Ethics Committee for Animal Management at the Shanghai Laboratory Animal Research Center, with approval number PA202300702. All mouse surgeries were performed by authors J.L. and X.W. at SLARC under approved protocols, in compliance with institutional policies and licensing requirements. Data collection and analyses for mice were performed by J.L., X.W., and Z.Z.

The adult male dog (Labrador, 3 years old) was used for the in vivo in-plane shielding validation experiments, and was provided by Harborside Medical Technology Company (Shanghai, China), a laboratory animal institution accredited by China National Accreditation Service for Conformity Assessment (CNAS LA0026), and the experimental protocol was approved by the Institutional Animal Care and Use Committee (IACUC) of this institution, with approval number IACUC-2023-019.

Five adult female sheep (Ovis aries, HuYang sheep, 1–2 years old, weighing 55-75 kg) used in this study were also obtained from and housed at the Harborside Medical Technology Company (Shanghai, China). All surgical and experimental protocols were approved by the IACUC of this institution (IACUC protocol 2023-090 and 2025-029) and were in accordance with The Association for Assessment and Accreditation of Laboratory Animal Care (AAALAC) International.

All dog and sheep surgeries were performed by licensed veterinarians at Harborside. The minimally invasive implantation workflow was designed by J.L., who also prepared the devices and assisted intraoperatively under veterinary supervision. Data collection and analyses for the dog and sheep were performed by J.L., L.S., and X.L.W. For acute and chronic sheep in vivo studies, ewes were consistently used for electrode implantation due to practical surgical considerations, which is a common approach in sheep electrophysiology. The sex of the mice and Labrador dog was not considered and was not associated with the findings and main conclusions of this study. Our experimental protocols and handling procedures for mice, dogs and sheep adhered to ethical standards for experimental animal research.

### In vivo validations on IVI
Male mice aged 10 weeks were used for the study to validate the effectiveness of the coplanar metal shielding layer in vivo. Induction anesthesia was administered using isoflurane, and anesthesia was maintained at 1–1.5% isoflurane concentration. A craniotomy was performed, and T0- and T1-types of IVI microelectrodes were attached to

the mouse cerebral cortex to record resting-state neural signals under anesthesia for a period. Epilepsy was induced through intraperitoneal injection of penicillin at a dose of 7 million U kg$^{-1}$. Following the injection of penicillin, the animals gradually exhibited rapid breathing. Throughout the surgery, electrophysiological monitoring of epileptiform discharges was conducted and recorded.

The dog was fasted for 12 h before anesthesia and re-fed after recovery. All procedures were aseptic under general anesthesia: intramuscular Zoletil®50 (Virbac, China, 6 mg/kg) for sedation and induction, endotracheal intubation with ventilatory support, and isoflurane in oxygen for maintenance. Physiological variables (heart and respiratory rates, blood pressure, end-tidal $CO_2$ levels, and vaporizer settings) were continuously monitored and documented at 15-min intervals, and crystalloid infusion was titrated according to standard operating procedures. After positioning and sterile draping, a temporal craniotomy (3.0 × 3.0 cm) was performed, and the bone flap was preserved on saline-soaked gauze. Following hemostasis, the dura mater was incised to expose the auditory cortex. For validation, T0- and T1-type IVI microelectrodes were gently placed subdurally on the cortical surface to acquire 10 min of acute electrocorticography. The IVI was then removed, and the scheduled clinical procedure proceeded. The dura and bone flap were re-approximated and secured, and the wound was closed in layers. Postoperative care followed institutional standards (including routine antibiotic prophylaxis). In vivo drug release validations of IVI were conducted on mice using the procedure from our previous work[45]. A triangular wound on the back of the mice was infected with *Staphylococcus aureus*, and the IVI in the experimental group was loaded with gentamicin.

## Establishment of sheep models of Parkinson's disease
A few days before the continuous administration, each ewe was placed with a jugular vein cannula under light anesthesia. To establish the chronic MPTP-induced Parkinsonian model, 1 mg/kg of MPTP hydrochloride (dissolved in 1 l of 0.9% sterile saline) was infused intravenously each day for 2 h, until signs of Parkinsonian symptoms such as unilateral neck deviation, limb rigidity, and reduced spontaneous activity were observed. Depending on the ewe's condition, the daily dosage could be escalated up to 2 mg/kg, with adjustments made to both the concentration and infusion rate as necessary. A total of five ewes underwent the chronic model establishment. The acute Parkinsonian sheep model was established through intraoperative intracerebral injections of carbachol (1.0 mM, dissolved in artificial CSF), targeting the dorsolateral striatum. In our experiments, we administered two separate injections of carbachol, first 5 µg and then 20 µg, sequentially.

## IVI implantation procedures
Ewes were food-deprived overnight prior to surgery. Anesthesia was induced using intravenous administration of Zoletil®50 (Virbac, China) at 6 mg/kg and maintained using 2–3% isoflurane through a Dräger medical ventilator. Throughout surgery, end-tidal $CO_2$ and mean arterial blood pressure were kept between 25–40 mmHg and 70–90 mmHg, respectively. Vital physical signs were recorded at regular time intervals throughout the procedure. A preoperative CT scan was conducted for the surgical path planning reference. In aseptic conditions, a midline incision was made from the nasion to the occipital bone to expose the dorsal surface of the skull. A retractor was inserted, and monopolar coagulation was employed for hemostasis. The attachment of the ewe's connective tissue to the cranial vault was relatively weak, necessitating the use of a bone scraper to remove any connective tissue, revealing a clear Bregma point. The Bregma point is situated at the intersection of the midline of the skull and the transverse suture between the frontal and parietal bones. All skull perforations and implantation targets were positioned by referencing the Bregma point as a bony landmark. A stainless-steel screw was screwed at the bregma as a ground electrode and CT marker, connected to the ground of the headstage by stainless

steel wire. The frontal sinus begins approximately 3.5 cm anterior and 2.5 cm lateral to the Bregma. The sensorimotor cortex is located directly beneath the frontal sinus. By performing skull perforations staying behind the line 35 mm anterior to the bregma, damage to the motor and somatosensory cortices can be significantly minimized (Supplementary Fig. 12). The ideal cortical puncture point is situated posterior to the postcruciate gyrus, approximately 20–25 mm anterior to Bregma. Additionally, it is crucial to avoid damaging the sagittal sinus, which marks the medial boundary of the implantation pathway. The stereotactic coordinate of the head of the caudate nucleus was determined intraoperatively using CT. Meanwhile, the determination of the craniotomy point coordinates and the planning of the implantation pathway were also finalized with intraoperative CT by measuring the relative distances and reference orientations between the implantation target and the skull screw markers.

The craniotomy was performed with an electric drill (6 mm) beginning 10 mm anterior to the bregma and 8 mm lateral to the midline. For the acute Parkinsonian model requiring intraoperative intracerebral injection, another 1-mm-diameter craniotomy was performed (centered 24 mm anterior to bregma, 5 mm lateral). After finishing the skull perforation procedure, the dura mater was incised using a dura incision tool. Flexible IVI encapsulated in a surgical catheter was implanted at the head of the caudate nucleus. A biocompatible silicone sealant (KWIK-CAST, World Precision Instruments, USA) was used to secure the surgical catheter and skull surface. During the operation, ewes received levodopa medication by subcutaneous injection of L-DOPA methyl ester/benserazide (95/25 mg/kg, soluble in 0.1% ascorbic acid solution, Aladdin, China). Chronic implantation was performed through a 4-mm diameter cranial burr hole. The backend of the IVI implant was sealed and secured using dental cement and Kwik-Sil biocompatible silicone. The skull-mounted base featured six anchoring flaps, each fastened with two M2 bone screws to ensure stable fixation. We successfully maintained stable anesthesia and electrophysiological setup for the duration of our studies, and the animals did not regain consciousness from anesthesia. Their euthanasia was performed at the conclusion of the study using intravenous administration of potassium. Death was confirmed by observing ventricular fibrillation followed by asystole on the Electrocardiograph readings. Post-mortem dissection and brain extraction, with visual inspection around the lateral ventricles, confirmed the successful deployment of the IVI on the head of the caudate nucleus.

## Immunohistochemistry procedures
At the end of the experiment, the ewes that had been euthanized were dissected to extract their brains, which were then fixed in 10% neutral buffered formalin for a week. Tissue sections and paraffin embedding were conducted on the substantia nigra, located between the dorsal tegmentum of the midbrain and the cerebral peduncle. Following dewaxing with xylene, the sections underwent a graded rehydration process through a series of ethanol solutions of descending concentrations, culminating in three successive PBS washes, each lasting 10 min. Antigen retrieval was achieved by agitating the sections in a 0.01 M citrate buffer (pH 6.0). Following deparaffinization and rehydration, the sections were rinsed three times with distilled water. They were then subjected to a peroxidase blocking solution for 10 min at room temperature (RT), followed by a five-times wash in PBS. A subsequent incubation with a protein blocking solution for 10 min at RT was performed. The sections were then exposed to a recombinant anti-tyrosine hydroxylase antibody (rabbit mAb) (targeting tyrosine hydroxylase, 1:500, Servicebio #GB15182-50, China) for 30 minutes at RT, followed by another five times washes in PBS. Incubation with a one-step HRP polymer for 30 minutes at RT preceded another series of washes—five in PBS and three in distilled water. Subsequently, the sections were treated with a few drops of ready-to-use diaminobenzidine reagent for 6–8 min at RT, then washed five times in PBS and three

times in distilled water. Hematoxylin solution was used for counterstaining. Tyrosine hydroxylase-positive cells appeared in deep brown. Three ewes were examined by immunohistochemistry. For immunofluorescence examinations, these primary antibodies were used: chicken anti-glial fibrillary acidic protein (GFAP) (targeting astrocytes, 1:1000, Abcam #ab4674, USA), goat anti-ionized calcium binding adapter molecule 1 (Iba 1) (targeting microglia, 1:500, Abcam #ab5076, USA), and rabbit anti-neuronal nuclear (NeuN) (targeting nuclei of neurons, 1:1000, Abcam #ab177487, USA).

Fluorescence image analysis was performed using ImageJ software (National Institutes of Health, USA). Neuronal and microglial cell densities were quantified through automated counting with the Particle Analyzer function. The same threshold intensity was applied to both the IVI-implanted and contralateral hemispheres to ensure consistency. Due to the network-like morphology of GFAP-labeled astrocytes, individual cell bodies could not be reliably identified. Therefore, astrocyte activation and accumulation were assessed by calculating the proportion of the area occupied by GFAP signals above a fixed intensity threshold. Cell densities (for neurons and microglia) and GFAP signal coverage (for astrocytes) from the implanted hemisphere were normalized to the corresponding average values from the contralateral control hemisphere.

### Neurophysiological data acquisition

Electrophysiological signals were acquired using a custom 32-channel headstage connected to a multichannel data acquisition system CereeCube NSP8 (Neuroxess Co., Ltd.), with a sampling rate of 4 kHz for all experiments. To mitigate the complex power frequency interference present in the CT operating room, repeated tests were conducted to improve the quality of intraoperative electrophysiological signals. The final setup was configured as follows: the I/O ground and chassis ground at the back of the RHD recording system were shorted; the skull screw was shorted with the shielding routing of the IVI, and with the ground on the headstage; the stainless-steel wire reference electrode was shorted with the reference on the headstage. During in vivo electrophysiological experiments on mice, the ground on the headstage was shorted with the stereotaxic instrument, and the reference on the headstage was shorted to a stainless-steel wire implanted under the skin tissue of the mouse's neck.

### Data analysis

The collected raw signals were preprocessed and analyzed by MATLAB (version R 2022b; Math Works, MA, USA), utilizing the EEGLAB and Fieldtrip toolboxes[46,47]. Initially, the signals were downsampled to 400 Hz, followed by notch filtering to remove powerline frequency interference and its harmonics. PSD analysis was conducted using the Welch's method. This analysis facilitated the screening and identification of peak frequencies within the beta frequency band. Time–frequency analysis and the spectrogram plotting were conducted using the pspectrum function in MATLAB. An epileptic mouse model was used to validate the in-plane shielded IVI. The SNR was determined by the decibel value of the RMS of the original signals during epileptic seizure periods compared to interictal periods[48]. Band-passed signals around the peak frequency were subjected to the Hilbert transform, and their magnitudes were taken to extract the envelope of the analytic signals (Supplementary Fig. 18). The analysis of beta burst activities was based on the envelope[49,50], where bursts are defined as durations of beta envelope that exceed the 80th percentile detection threshold with a minimum duration of 0.1 s.

### Neural activity discrimination

The same processing methods as beta burst analysis were used to obtain the Hilbert envelope. Then, dimensionality reduction was performed using SVD, extracting the first two principal components for subsequent LDA, thereby constructing an SVD–LDA classification model. Effective data spanning 10 min each from three neural activity states (chronic Parkinsonian state, pathological beta burst state, post-levodopa administration state) were put in the model, with the dataset divided into 60% for training and 40% for testing. Within the training set, a 10-fold cross-validation was employed to establish the optimal model, and the model's generalization performance in terms of discrimination accuracy was validated on the test set. All discrimination accuracy results were obtained after conducting training and validation on datasets that were randomly divided 30 times.

### Statistics

Data are expressed as mean ± standard deviation (SD) unless stated otherwise. For statistical tests involving line noise, SNR, and beta bursts features under different experimental conditions, unpaired two-tailed Student's $t$-test or Mann−Whitney tests were used. When analyzing differences in burst duration of several windows, unpaired multiple Mann-Whitney tests were used, followed by Bonferroni−Dunn corrected tests. When comparing noises of different IVI microelectrodes, way analysis of variance was used, followed by Bonferroni-corrected unpaired $t$-tests. The models were constructed using Prism (GraphPad Software, Version 9). All confidence intervals reported are 95% confidence intervals. Statistical significance was accepted as: *$p < 0.033$, **$p < 0.002$ and ***$p < 0.001$.

### Reporting summary

Further information on research design is available in the Nature Portfolio Reporting Summary linked to this article.

## Data availability

The source data generated in this study are provided in the Source Data.xls file. The raw data for electrophysiology used for the analysis in this study are available from Zenodo (https://doi.org/10.5281/zenodo.16936239). Source data are provided with this paper.

## Code availability

The custom MATLAB scripts used for electrophysiology analysis are available from Zenodo (https://doi.org/10.5281/zenodo.16936239).

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

## Acknowledgements

We thank F. Xu for helping with experiment preparation and Z. Zhu for helpful discussion on the silk scaffold self-unfolding mechanism. This work was partially supported by National Key R & D Program of China (Grant No. 2019YFA0905200, X.L.W.), Youth Innovation Promotion Association for Excellent Members, CAS (Grant No. Y2023070, Z.Z.), Shanghai Rising-Star Program (Grant No. 22QA1410900, Z.Z.), Key Research Program of Frontier Sciences, CAS (Grant No. ZDBS-LY-JSC024, T.H.T.), Shanghai Pilot Program for Basic Research-Chinese Academy of Science, Shanghai Branch (Grant No. JCYJ-SHFY-2022-01, T.H.T.), Shanghai Municipal Science and Technology Major Project (Grant No. 2021SHZDZX, L.S.), The Jiangxi Province 03 Special Project and 5 G Project (Grant No. 20212ABC03W07, Z.Z.).

## Author contributions

J.L. and X.W. contributed equally to this work. Z.Z., T.H.T., Z.S., and J.L. conceived the idea. Z.Z., J.L., X.W., and T.H.T. designed the experiments. Z.C. fabricated the IVI microelectrodes. K.L. fabricated the silk wafers. J.L., X.W., Z.Z., L.S., and X.L.W. performed the experiments. J.L., Z.Z., and X.W. analyzed the data. J.L., Z.Z., and T.H.T. wrote the paper. All authors discussed the results and provided comments for the paper.

## Competing interests

T.H.T. is a founder of Neuroxess Co., Ltd., and Z.C. is employed by the company. Neuroxess may in the future use our intraventricular interfaces for research or commercial activities derived from this work. The remaining authors declare no competing interests.
