## [Transparent Peer review file · Nature Communications]

Silk-enabled Conformal Intraventricular Interfaces for Minimally Invasive Neural Recordings

Corresponding Author: Professor Zhitao Zhou

Version 0:

Reviewer comments:

Reviewer #1

(Remarks to the Author)

In this manuscript, Liang et al. reported a self-unfolding and conformal microelectrode array paired with a silk scaffold, which can be targeted into deep brain regions. The authors exploited the shape memory polymers as self-unfolding actuator, and demonstrated efficient delivery through surgical catheter to the target brain regions. In-plane shielding design proposed into the DBNI is valuable for neural recording in the complicated operating room environment. Recordings in the sheep demonstrated the capabilities of dMEA in the neural detection of deep brain regions. However, some questions should be addressed before consideration for publication:

1. The concern is that the size of the surgical catheters with an inner diameter of 5 mm is quite large and harmful for the brain regions passed by the implantation trajectory, which may limit the clinical translation of this technique. Moreover, the diameter of the catheters may depend on the space of dMEA center part and the size of FPC. Further applications remain to be considered.
2. As seen from the Supplementary Video 1, the reflow soldering pads connected with FPC could contact with fluid (CSF for in vivo experiments) which may damage the electronic circuit. Stable encapsulation is crucial for the neural implants. The authors should provide more details about the connection and packaging between dMEA and FPC to prevent the CSF erosion.
3. As seen from the Supplementary Video 1, a brief segment of FPC was protruded from the catheter, which may be non-biocompatible. Nickel and tantalum were also used in the DBNI. The authors should demonstrate the biocompatibility of the DBNI for neural tissues.
4. A suggestion about the description of n value: for example, in the legends of Fig 2E and 2F, it would be beneficial to describe as “n = m1 electrode sites/channels from m2 devices”.

Reviewer #2

(Remarks to the Author)

Current recording devices directed at subcortical structures are geometrically limited such that they can typically record along a single trajectory, but cannot easily be adapted to record over a larger volume or area of deep brain nuclei. This work sought to develop a recording array that could be used to target the CSF-facing surfaces of deep brain structures to record over a greater extent than is otherwise typically possible.

The “addressable” brain using this technique is relatively limited (the portions of deep brain structures abutting CSF spaces), and there are no current, established indications for those potential targets, and relatively few experimental strategies. Even for the few conceivable experimental examples (e.g., the demonstration of PD-model-related beta activity changes in the caudate), there is no current evidence or even rationale to strongly support a contention that this approach would be superior therapeutically.

Notably, “brush-style” electrodes that extend from a central cannula and allow the recording of a volume of tissue within a deep brain structure do exist, and these do have more “addressable” volume, so this approach is the only method that enables interfacing with deep neural structures over an extended, non-co-axial volume.

Whether the drug-release capability is intended to be simply for antibiotic prophylaxis or if there is some intended additional / alternative use is unclear. Scenarios that would benefit beyond antibiotic prophylaxis are difficult to imagine, and would be fairly contrived at this time.

The critical factor of the stability of the array and the array-neural interface over time and in the setting of CSF pulsations is unknown.

Despite being labeled as a "deep brain" neural interface, some experiments were conducted in the subdural space (presumably over the cortical convexity) in mice and dogs; whether this is an intended potential use of the device is unclear, though the notion that one could minimally-invasively implant unfolding electrocorticography arrays over the lateral cortex could be appealing if feasible. Unfortunately, these in vivo experiments appeared to require a craniotomy, so one can surmise that the subdural space likely wouldn't accommodate the unfolding structure otherwise.

Overall, while the engineering appears innovative, given the present state of our understanding of the neural pathways associated with neurological and psychiatric diseases, and the state of circuit modulation therapies, this device appears to be a solution in search of a problem. Importantly, whether such a device could indeed function chronically in light of the mechanical (fluid dynamic) and biological (tissue reaction at the ependyma) forces that will be encountered over time is unknown.

Version 2:

Reviewer comments:

Reviewer #1

(Remarks to the Author)

The authors have fully addressed all the questions. I recommend the publication of this manuscript.

Reviewer #2

(Remarks to the Author)

This significantly improved manuscript demonstrates the feasibility of an intraventricular recording device, chronically implanted in sheep in the context of a Parkinson's disease model. The demonstration of a beta-power peak in the electrophysiological signals that is responsive to levodopa suggests that such an approach can in principle detect relevant disease biomarkers. While the extent of the "addressable" brain remains relatively limited (subcortical structures abutting the ventricle with sufficient ventricular volume locally to allow the device to unfold), and while there are no current indications for such an interface (even in Parkinson's, where activity in the non-ventricular-facing putamen rather than caudate is more relevant to motor symptoms that are currently treatable with deep brain stimulation), this is nonetheless a creative and interesting approach to expand the range of options available for interfacing with various brain structures, even if it is, at the current moment, a solution in search of a problem.

Point by point response (comments in black and responses in blue):

Reviewer #1:

[1] In this manuscript, Liang et al. reported a self-unfolding and conformal microelectrode array paired with a silk scaffold, which can be targeted into deep brain regions. The authors exploited the shape memory polymers as self-unfolding actuator, and demonstrated efficient delivery through surgical catheter to the target brain regions. In-plane shielding design proposed into the DBNI is valuable for neural recording in the complicated operating room environment. Recordings in the sheep demonstrated the capabilities of dMEA in the neural detection of deep brain regions. However, some questions should be addressed before consideration for publication.

We sincerely thank the reviewer for the insightful summary and comments on our work and we have carefully addressed all the points raised during the revision. Furthermore, to enhance terminological precision in this work, we have also revised the original designation from "deep brain neural interfaces (DBNI)" to "Intraventricular Interfaces (IVI)".

[2] The concern is that the size of the surgical catheters with an inner diameter of 5 mm is quite large and harmful for the brain regions passed by the implantation trajectory, which may limit the clinical translation of this technique. Moreover, the diameter of the catheters may depend on the space of dMEA center part and the size of FPC. Further applications remain to be considered.

We thank the reviewer for raising the critical concern regarding the 5-mm inner diameter surgical catheter's potential impact on implantation trajectory brain regions and its clinical translation implications. We prioritized the anatomical constraint during large animal experiment design. The sheep model was selected specifically for its neuroanatomical similarity to human deep brain structures, particularly the basal ganglia network central to Parkinson's disease research. Given that the complexity of skull and frontal sinus structure and difficulty of craniotomy in sheep, we implemented a 6-mm craniotomy drill paired with the 5-mm catheter for prototype validation in this model. During preoperative planning for the Parkinson's disease sheep model, the trajectory was carefully selected to avoid critical brain regions involved in motor function.

Regarding catheter scalability, our current device specifications (center part of dMEA: 2.32 mm × 2.28 mm, the width of FPC: 2.31 mm) demonstrate theoretical compatibility with clinical external ventricular drainage catheters (O.D.: 4.9 mm, I.D.: 2.6mm). Through flip-chip bonding (Finetech GmbH & Co. KG), achieving 160- μ m pad pitch precision, and multi-layer FPC design optimization, the device dimensions can be reduced. For enhanced clinical viability, we developed miniaturized components (center part of dMEA: 1.27 × 1.27 mm; the width of FPC: 1.45 mm) compatible with commonly used 2.2-mm neurosurgical catheters (**Figure R1a-d/New Supplementary Fig. 3a-d**). We conducted electrical characterization targeting this small-sized dMEA and the IVI as prepared. Following the assembly procedure described in the original manuscript, the IVI exhibited no significant changes in its electrical performance and maintained stable impedance, as shown in **Figure R1e/New Supplementary Fig.3e**.

Moreover, we conducted chronic *in vivo* studies using the fabricated small-sized intraventricular neural interface, achieving stable implantation into the sheep's lateral ventricle through a minimally invasive 4 mm-diameter burr hole, demonstrating translational potential for clinical applications. To ensure stable long-term recordings, a skull-mounted base fabricated from high-strength 3D-printed nylon was designed to protect the backend of the IVI. The complete IVI

system and its implantation target are shown in **Figure R2/New Fig. 1D-E**. Stable electrophysiological recordings near the caudate nucleus in freely moving sheep were maintained for over 4 weeks, providing preliminary evidence for clinical translation (**Figure R3/New Fig. 5A-B**). We again thank the reviewer for the valuable feedback that helped clarify the limitation of this technique and improve our work.

Figure R1. Small-sized intraventricular Interfaces (IVIs). (a) The layout design of the small-sized deformable microelectrode array. (b) The corresponding small size flexible printed circuit board. (c) The image of small size IVI device. (d) The image of small size IVI integrated into medical catheter with inner diameter of 2.2 mm. (e) Impedance tests of IVI before and after the catheter assembly (n = 48 electrode sites from 3 devices).

Figure R2. The complete IVI system and its implantation scenario. (A) The photo of the

complete IVI system including the implant and the skull-mounted base. (B) The image of unfolded state of the implant device including the small-sized dMEA and the corresponding silk scaffold. Upper image shows the caudate nucleus head being pointed as implantation target.

Figure R3. Intraventricular electrophysiology in freely moving sheep. (A) The photo of the free-moving sheep after surgery. During post-operative care and recovery, the implant and surgical incision are further protected with a pedestal and an Elizabethan collar. (B) The waveforms of neural activity recorded by the IVI on free-moving sheep during four weeks.

[3] As seen from the Supplementary Video 1, the reflow soldering pads connected with FPC could contact with fluid (CSF for in vivo experiments) which may damage the electronic circuit. Stable encapsulation is crucial for the neural implants. The authors should provide more details about the connection and packaging between dMEA and FPC to prevent the CSF erosion.

We sincerely appreciate the reviewer's insightful suggestions and regret the oversight in methodological clarification in the original manuscript. The reflow soldering pads are encapsulated with a biocompatible ultraviolet-curable adhesive (MD@1128A-M-VT, DYMAX, USA) [Oxley, T. J. et al. *Nat. Biotechnol.* **34**, 320–327 (2016)], as shown in Figure R4a/New Supplementary Fig. 8a. Recordings conducted over several hours confirmed no instability or performance degradation under CSF exposure, with this duration being compatible with that of typical external ventricular drainage surgery. Given that neurosurgical procedures may require temporary implant retention for postoperative monitoring [Chen, C. et al. *PLoS ONE* **9**, e101961 (2014); Lu, P. et al. *Front. Neurol.* **10**, (2019)], electrical impedance tests were performed during a four-week period following dMEA being immersed in artificial CSF at 37 °C. The stable impedance profiles confirmed successful encapsulation between dMEA and FPC (Figure R4b/New Supplementary Fig. 8b). The methodological details have been comprehensively clarified in Methods section of the revised manuscript.

Figure R4. Packaging of the dMEA. (a) The image of the ultra-violet curable adhesive encapsulation of the dMEA and the FPC. (b) Stable encapsulation in artificial CSF at 37°C enduring 4 weeks verified by impedance spectrum tests at 1st week, 2nd week and 4th week after the immersion (n = 24 electrode sites).

[4] As seen from the Supplementary Video 1, a brief segment of FPC was protruded from the catheter, which may be non-biocompatible. Nickel and tantalum were also used in the DBNI. The authors should demonstrate the biocompatibility of the DBNI for neural tissues.

We sincerely thank the reviewer for the valuable comments. The protruding FPC segment observed in Supplementary Video 1 is also encapsulated with the same biocompatible adhesive. In this design, nickel is exclusively employed in the soldering layer (Cr/Ni/Au) of the reflow pads, which is entirely encapsulated within the biocompatible adhesive, ensuring secure sealing during operation. The tantalum powder blended into the biocompatible adhesive functions solely as a CT contrast agent without intraoperative leakage. Tantalum's biocompatibility is well-established [Li, M. et al. *Bioact. Mater.* **11**, 140–153 (2022); Wang, X. et al. *J. Mater. Res. Technol.* **30**, 1706–1715 (2024)], and the neural tissue biocompatibility of dMEA and silk scaffolds has also been validated in prior work [Zhou, Y. et al. *Microsyst. Nanoeng.* **8**, 1–12 (2022); Liu, K. et al. *Adv. Healthc. Mater.* **7**, 1701359 (2018)].

Third-party residue analysis involved electrode immersion in 37°C (human body temperature) ultrapure water for 30 days, with subsequent leaching tests revealing no detectable fabrication-related compounds (**Figure R5**).

[FIGURE REDACTED]

Figure R5. Test report on residues in electrode leaching solution after 30 days.

Neural tissue compatibility was assessed via epidural implantation in mice of all contact components (dMEA, silk scaffold, tantalum marker, and FPC soldering pad parts), followed by immunohistochemistry studies, as shown in **Figure R6/New Supplementary Fig. 4**. Results at four

weeks post-implantation indicate good biocompatibility of the device. While transient increases in astrocytes and microglia were detected at the implantation site one week post-operation, this inflammatory response aligns with established neural interface paradigms. Immunohistochemical profiling of analogous devices consistently demonstrates such glial activation during the initial 1-2 week recovery phase, as evidenced by prior studies [Wei, S. et al. *Nat. Commun.* 15, 715 (2024)].

Figure R6. Biocompatibility of IVI for neural tissues. (a) Immunohistochemistry studies on mice with the IVI implanted epidurally. Immunofluorescence images of a mice cortex under the IVI (implanted) and on the contralateral control side (control) at different time points after implantation. Scale bar, 100 μm . (b) Normalized cell density of NeuN-labelled neurons. (c) Normalized signal of GFAP-labelled astrocytes. (d) Normalized cell density of Iba1-labelled microglial cells. The box plots show the median and quartile range. $n = 16$ from 3 mouse for week 1 and 4 data.

To further evaluate chronic biocompatibility, terminal immunofluorescence analysis was performed at the endpoint of four-week implantation studies. Quantitative comparisons of NeuN-labelled neurons, Iba1-labelled microglia, and GFAP-labelled astrocytes between the IVI-implanted and contralateral hemispheres in coronal sections demonstrated absence of significant immune rejection (**Figure R7/New Fig. 5J**). We appreciate the insightful comment and have addressed this in the revised manuscript.

Figure R7. Long-term biocompatibility of IVI on the parkinsonian sheep model. Immunofluorescence images showing the periventricular structures on the IVI-implanted and contralateral control sides of the sheep brain. Quantitative analysis included the normalized cell density of NeuN-labelled neurons ($p = 0.22$), normalized cell density of Iba1-labelled microglial cells ($p = 0.65$) and normalized signal of GFAP-labelled astrocytes ($p = 0.83$). Two-tailed paired t tests were used.

[5] A suggestion about the description of n value: for example, in the legends of Fig 2E and 2F, it would be beneficial to describe as “n = m1 electrode sites/channels from m2 devices”.

We thank the reviewer for the valuable suggestion. Supplementary tests and amendments information have been incorporated into the corresponding sections of the revised manuscript, as shown in **Figure R8/ Fig. 2E-F**. For the Fig R5a and Fig R5b, n = 48 electrode sites from 3 devices (Each device has 16 electrode sites). In the revised manuscript, similar clarifications regarding the n value have also been duly addressed.

Figure R8. (a) The direct current resistance fluctuations of DBNI microelectrodes (n = 48 electrode sites from 3 devices) arising from 100 cycles of bending and warping. (b) Impedance curves of the DBNI microelectrodes before and after the assembly (n = 48 electrode sites from 3 devices).

Reviewer #2:

[1] Current recording devices directed at subcortical structures are geometrically limited such that they can typically record along a single trajectory, but cannot easily be adapted to record over a larger volume or area of deep brain nuclei. This work sought to develop a recording array that could be used to target the CSF-facing surfaces of deep brain structures to record over a greater extent than is otherwise typically possible.

We sincerely thank the reviewer for the positive comments and summary of the characteristics of our work. In the revised manuscript, we have conducted additional *in vitro* and *in vivo* evaluations to further validate the chronic safety, functionality, and stability of the technology, thereby strengthening its translational potential.

[2] The “addressable” brain using this technique is relatively limited (the portions of deep brain structures abutting CSF spaces), and there are no current, established indications for those potential targets, and relatively few experimental strategies. Even for the few conceivable experimental examples (e.g., the demonstration of PD-model-related beta activity changes in the caudate), there is no current evidence or even rationale to strongly support a contention that this approach would be superior therapeutically.

We thank the reviewer for the insightful comment. We acknowledge the observation regarding the current limitations in “addressable” brain regions. While this technique indeed focuses on CSF-adjacent periventricular structures, it uniquely enables conformal neural recordings unattainable through existing methods like SEEG, brush-style electrodes, or endocisternal interfaces [Steinmetz, N. A. et al. *Science* 372, eabf4588 (2021); Zhao, Z. et al. *Nat. Biomed. Eng.* 7, 520–532 (2023); Liu, Y. et al. *Nat. Neurosci.* 27, 1620–1631 (2024); Chen, J. C. et al. *Nat. Biomed. Eng.* 1–9 (2024)]. Crucially, the catheter-deployable design allows access to all periventricular neural surfaces, providing a platform to monitor surface electrophysiological signals across these understudied regions.

The physiological significance of CSF-adjacent deep brain structures has been reported. The ventricular system plays a crucial role in neurogenesis, neurodegeneration, CSF circulation, and psychiatric disorders. Neural stem cells along the ependyma serve as key sources for brain development. Abnormal ventricular morphology and associated electrophysiological signatures may reflect neurodevelopmental deficits and serve as potential biomarkers for neurological and psychiatric diseases, such as Alzheimer’s disease, autism and hydrocephalus [Fame, R. M., Corté s-Campos, C. & Sive, H. L. *BioEssays* 42, 1900186 (2020); Duy, P. Q. et al. *Neuron* 110, 12–15 (2022); Ge, Y.-J. et al. *Nat. Hum. Behav.* 8, 164–180 (2024)]. Our work, therefore, provides an essential tool for exploring these regions, opening new avenues for experimental strategies. Building upon this argument, we have changed the title of the manuscript to “Silk-enabled Conformal Intraventricular Interfaces for Minimally Invasive Neural Recordings” to better reflect the original intent and practical scope of the technology’s development. And we have referred to this technology as “Intraventricular Interfaces (IVI)” in place of the original term “deep brain neural interfaces (DBNI)”, as well as in the revised manuscript.

Researches on the caudate nucleus have long been a central focus in the study of basal ganglia–cortical neural circuits and the pathology of related psychiatric disorders [Grahn, J. A., Parkinson, J. A. & Owen, A. M. *Prog. Neurobiol.* 86, 141–155 (2008); Macpherson, T. & Hikida, T.

Psychiat. Clin. Neuros. 73, 289–301 (2019); **Banwinkler, M., Theis, H., Prange, S. & van Eimeren, T. Brain Sci.** 12, 1248 (2022)]. Morphometric studies of striatal structures also suggest that further investigation into the electrophysiology of striatal subregions is of significant value for identifying diagnostic biomarkers [**Khan, A. R. et al. NeuroImage-Clin.** 21, 101597 (2019)]. In recent years, there have also been studies exploring the caudate nucleus as a novel target for neuromodulation and stimulation therapies [**Munoz, F. et al. Brain Stimul.** 15, 360–372 (2022); **Conde-Berriozabal, S. et al. Exp. Neurol.** 383, 114991 (2025)].

Our work is not intended to directly provide a more effective treatment method. Instead, it aims to offer a new tool for the monitoring of periventricular nuclei surface electrophysiology to record potential pathological electrophysiological markers. Moreover, it could serve to explore the possibility of more effective and more natural therapeutic strategies. This tool can be used to study and provide electrophysiological signals from the surface of a wide range of deep brain nuclei. For example, in the case of DBS, current sEEG stimulation therapy is superior in alleviating Parkinson's symptoms. Still, researchers are exploring the neural mechanisms of closed-loop regulation in DBS and searching for more natural stimulation parameters. We envision that this technology will enable researchers to chronically monitor periventricular neural circuits and to identify biologically relevant stimulation targets and parameters. Such advances could open new avenues for real-time monitoring, early warning, and therapeutic intervention in neurological and psychiatric disorders involving periventricular brain structures. Thanks again for the insightful comment and we have clarified it in our revised manuscript.

[3] Notably, “brush-style” electrodes that extend from a central cannula and allow the recording of a volume of tissue within a deep brain structure do exist, and these do have more “addressable” volume, so this approach is the only method that enables interfacing with deep neural structures over an extended, non-co-axial volume.

We thank the reviewer for raising this important issue and apologize for any lack of clarity in our original manuscript. We fully agree that “brush-style” electrodes achieve greater addressable volume for recording within deep brain tissue. In fact, the original aim of our technology was not to record electrical signals from the internal volume of widely accessible deep brain structures, which would inevitably involve invasive damage to the targeted neural nuclei. Instead, this technique was specifically designed to better adapted to the periventricular neural structures and to record large-scale surface electrophysiological signals from deep brain regions within the CSF milieu, which is beyond the capability of macro- and micro-electrodes, including “brush-style” electrodes [**Jiang, S. et al. Nat. Commun.** 11, 6115 (2020); **Liu, Y. et al. Nat. Neurosci.** 27, 1620–1631 (2024)]. We have also clarified in the revised text, this approach prioritizes minimally invasive access to periventricular electrophysiology through adaptive structural design, rather than competing with existing volumetric recording modalities. We appreciate this opportunity to better articulate our technical positioning.

[4] Whether the drug-release capability is intended to be simply for antibiotic prophylaxis or if there is some intended additional / alternative use is unclear. Scenarios that would benefit beyond antibiotic prophylaxis are difficult to imagine, and would be fairly contrived at this time.

We thank the reviewer for the insightful comment. The manuscript demonstrates drug-release

functionality specifically for antibiotic delivery. In neural interface applications, electrode coatings or substrates are extensively employed to enable in situ anti-inflammatory drug delivery, effectively mitigating post-implantation neural inflammation and immune rejection [Green, R. & Abidian, M. R. *Adv. Mater.* **27**, 7620–7637 (2015); Hu, Z., Niu, Q., Hsiao, B. S., Yao, X. & Zhang, Y. *Mater. Horiz.* **10**, 808–828 (2023)]. In the absence of targeted intervention, immune responses in neural tissue may frequently lead to the formation of scar tissue encapsulating the flexible electrodes, ultimately compromising their long-term recording performance. This technology could naturally be adapted for this purpose. Additionally, delivering anti-inflammatory drugs via the neural electrode during the recovery period is also a considered strategy.

Therefore, from the initial design of the IVI, we prioritized the capability for localized drug delivery to enhance resistance against neuroimmune rejection and to meet the practical demands of therapeutic delivery in clinical applications. Our previous work has already demonstrated the advantages of silk fibroin materials as substrates or coatings for drug loading and in situ controllable release [Zhang, S. et al. *Adv. Sci.* **7**, 1903802 (2020); Sun, L. et al. *Small* **16**, 2000294 (2020)]. In addition to loading antibiotics, silk fibroin functional materials can also carry phenobarbital, catalase, as well as blood biomarkers such as hemoglobin, albumin, and glucose [Zhou, Z. et al. *Adv. Mater.* **29**, 1605471 (2017); Shi, Z. et al. *Adv. Sci.* **6**, 1801617 (2019); Lee, W. et al. *Nat. Nanotechnol.* **15**, 941–947 (2020)]. The introduction of silk fibroin, which is biocompatible, biodegradable, and has tunable mechanical properties, as a bioactive drug-loaded substrate not only improves the overall biocompatibility of the IVI but also broadens its potential applications in minimally invasive implantation. Thanks again for the valuable feedback and we have clarified it in our revised manuscript.

[5] The critical factor of the stability of the array and the array-neural interface over time and in the setting of CSF pulsations is unknown.

We thank the reviewer for the valuable comment, and we are sorry for not clarifying this in the original manuscript. First, regarding the stability of the array over time and in the setting of CSF pulsations, the encapsulation strategy and material selections of this technology have been thoroughly evaluated to ensure the reliability of stable recordings for up to four weeks, which is comparable with the typical duration of intracranial sEEG electrode implantations. Given that neurosurgical procedures may require leaving the implant in place for a few days or even four weeks for postoperative care, and in order to investigate critical factor that may influence its stability over a relatively longer period, we conducted the following *in vitro* and *in vivo* experiments to evaluate the stability of the array-neural interface over time and in the setting of CSF pulsations.

The reflow soldering pads are encapsulated with a biocompatible ultraviolet-curable adhesive (MD®1128A-M-VT, DYMAX, USA) [Oxley, T. J. et al. *Nat. Biotechnol.* **34**, 320–327 (2016)], as shown in **Figure R9a/New Supplementary Fig. 8a**. Recordings conducted over several hours confirmed no instability or performance degradation under CSF erosion. And for further validations of potential prolonged use, electrical impedance tests were conducted during the four-weeks window after dMEA being immersed in artificial CSF at 37°C, and the stable impedance property confirmed successful encapsulation between dMEA and FPC (**Figure R9b/New Supplementary Fig. 8b**). On the other hand, since the CSF pulsation frequency is synchronized with the cardiac cycle [Beltrán, S. et al. *NMR Biomed.* **37**, e5013 (2024)], any noise introduced by the CSF

pulsation is unlikely to affect the brain wave of interest frequencies, as well as higher-frequency electrophysiological signals.

Figure R9. Packaging of the dMEA. (a) The image of the ultra-violet curable adhesive encapsulation of the dMEA and the FPC. (b) Stable encapsulation in artificial CSF at 37°C enduring 4 weeks verified by impedance spectrum tests at 1st week, 2nd week and 4th week after the immersion (n = 24 electrode sites).

Furthermore, we extended our evaluation to an *in vivo* chronic sheep model to assess the stability of the IVI within a physiologically pulsatile CSF environment. Horizontal CT slices acquired intra-operatively and at two postoperative follow-ups revealed no significant micromotion of the IVI relative to the lateral ventricle (**Figure R10/New Fig.5I**). In addition, continuous impedance measurements of channel impedance demonstrated no deterioration in the electrical performance of the IVI over time (**Figure R11a/New Supplementary Fig. 25a**). The signal-to-noise ratio (SNR) was continuously monitored, and the results confirmed that there was no substantial change or degradation in the device's functional stability (**Figure R11b/New Supplementary Fig. 25b**). Thank again for raising this important concern and we have included these results and corresponding discussion into the revised manuscript to further substantiate the reliability and long-term stability of our array.

Figure R10. Intraoperative and follow-up CT scans demonstrated the restoration of lateral ventricle morphology and the mechanical stability of the IVI, with no apparent displacement.

Figure R11. The fluctuation of the IVI channels' impedance and signal to noise ratio (SNR) during *in vivo* chronic experiment period.

[6] Despite being labeled as a “deep brain” neural interface, some experiments were conducted in the subdural space (presumably over the cortical convexity) in mice and dogs; whether this is an intended potential use of the device is unclear, though the notion that one could minimally-invasively implant unfolding electrocorticography arrays over the lateral cortex could be appealing if feasible. Unfortunately, these *in vivo* experiments appeared to require a craniotomy, so one can surmise that the subdural space likely wouldn't accommodate the unfolding structure otherwise.

We thank the reviewer for highlighting this issue and sincerely regret any ambiguity in the original manuscript. The experiments on mice and dogs were conducted to validate the *in vivo* effectiveness of the coplanar metal shielding layer, thereby complementing the conclusions from *in vitro* testing. Notably, the unfolding implantation of electrocorticography arrays lies beyond the scope of this technique, though preliminary studies in our lab are exploring shape memory techniques in related applications. This technology is designed to capitalize on established neurosurgical techniques enabling ventricular access, providing a minimally invasive method for recording electrophysiological signals from the surfaces of periventricular neural tissues adjacent to the cerebrospinal fluid. Furthermore, the ball-to-plane transformation mechanism of the IVI is incompatible with epidural unfoldment requirements. We appreciate this valuable feedback and have incorporated these clarifications in the revised manuscript.

[7] Overall, while the engineering appears innovative, given the present state of our understanding of the neural pathways associated with neurological and psychiatric diseases, and the state of circuit modulation therapies, this device appears to be a solution in search of a problem. Importantly, whether such a device could indeed function chronically in light of the mechanical (fluid dynamic) and biological (tissue reaction at the ependyma) forces that will be encountered over time is unknown.

We thank the reviewer for the insightful comment and forward-looking suggestion. We would like to emphasize that the primary goal of this technology is to leverage established neurosurgical paradigms and existing indications that allow access to the ventricles, providing a minimally invasive strategy for intraoperative and chronic recording of electrophysiological signals from the surface of periventricular neural structures adjacent to the CSF. This approach avoids the need for

additional surgical procedures or operative routes for electrophysiological monitoring, ensuring safety comparable to that of clinical electrocorticography (ECoG) use at this point. The potential research significance of monitoring periventricular neural activity has been gradually recognized [Duy, P. Q. et al. *Neuron* 110, 12–15 (2022); Ge, Y.-J. et al. *Nat. Hum. Behav.* 8, 164–180 (2024)]. Intraoperative and chronic clinical applications could provide valuable insights into pathological electrophysiological characteristics of periventricular neural structures, potentially offering new perspectives for studying neural regeneration, functional changes in neural networks following neuronal differentiation, and the pathophysiology of brain disorders such as Alzheimer's disease, autism, and hydrocephalus.

To further evaluate the translational potential of this technology, we have performed both *in vitro* and *in vivo* studies assessing the chronic safety, functionality, and stability of the small-sized IVI devices (Figure R1/New Supplementary Fig. 3, Figure R2/New Fig. 1D-E). The aforementioned *in vitro* experiments provide preliminary evidence regarding the mechanical stability of this neural interface. Specifically, the four-week impedance testing in phosphate-buffered saline confirmed stable encapsulation between the dMEA and FPC (Figure R9/New Supplementary Fig. 8). Furthermore, the agarose model experiments demonstrated the dynamic adaptability of the interface and its ability to maintain stable, conformal attachment to curved surfaces with the assistance of a silk scaffold (Figure R12/New Supplementary Fig. 6). These findings support the device's capability to maintain stability within the pulsatile CSF environment.

Figure R12. *In vitro* self-unfolding and conformal attachment validations of IVI on agarose brain fantom model with different curvatures.

As for chronic *in vivo* studies, we successfully conducted a four-week experiment using the small-sized IVI in a parkinsonian sheep model, enabling chronic recordings of neural activity near the caudate nucleus across the progression from pre- to post-Parkinsonian states. These studies aimed to meet clinical needs for postoperative monitoring and neuroscientific demands for long-term tracking of periventricular neural dynamics during disease progression. The recording period

was aligned with the clinical duration of sEEG electrode use (**Figure R13/New Supplementary Fig. 24**). We assessed the chronic functionality from both mechanical and biological perspectives (**Figure R14/New Fig. 5**). Specifically, the PD sheep model was induced using intravenous infusion of MPTP during the second week after implantation. We found a beta peak at 18 Hz during recordings at the fourth week, and the further beta burst analysis was conducted (**Figure R14a-d**). The windowed distribution analysis of beta burst durations clearly revealed differences in beta burst characteristics before and after PD induction, as well as a consistent evolution throughout the PD modeling process. Further statistical analysis of burst duration, amplitude, and frequency demonstrated a progressive emergence of beta bursts with greater amplitude, longer duration, and higher frequency during the course of chronic PD model development (**Figure R14e-h**). Mechanical stability was assessed by comparing intraoperative and postoperative follow-up CT scans (**Figure 14i**), which revealed no evident displacement of the IVI relative to the lateral ventricle and a gradual restoration of ventricular morphology. For long-term biocompatibility, terminal immunofluorescence analysis was performed at the endpoint of four-week implantation studies. Quantitative comparisons of NeuN-labelled neurons, Iba1-labelled microglia, and GFAP-labelled astrocytes between the IVI-implanted side and contralateral side in coronal sections demonstrated absence of significant immune rejection after four weeks of implantation (**Figure R14j**). Our results demonstrate that the small-sized IVI serves as a reliable neural interface for chronic recording of intraventricular electrophysiological dynamics, revealing distinct transitions between healthy and pathological brain states and their responsiveness to external stimuli. Thanks again for the insightful comment, which has helped us to clarify this important aspect of our work.

Figure R13. Experiment scheme of the *in vivo* chronic recordings in parkinsonian sheep model.

Figure R14. Intraventricular electrophysiology in freely moving sheep. (a) The photo of the free-moving sheep after surgery. During post-operative care and recovery, the implant and surgical incision are further protected with a pedestal and an Elizabethan collar. (b) The waveforms of neural activity recorded by the IVI on free-moving sheep during four weeks. (c) Power spectral density of local field potentials recordings (mean \pm SD) ($n = 5$), with a beta peak at 18 Hz during recordings at the fourth week. (d) Beta burst amplitude and duration were extracted from five-minute artifact-free recordings obtained during each of the three phases: post-operative, MPTP modeling, and the Parkinsonian state. (e) Bar plots showing changes in distribution of burst durations as a percentage of total number of bursts (mean + SEM), during these three phases. 2way ANOVA tests: 0.1-0.15 s, $p < 0.001$; 0.15-0.2 s, $p = 0.04$; 0.3-0.4 s, $p = 0.009$. (f-h) Unpaired t tests of beta bursts characteristics. F: The duration of all bursts during three phases, $F(2, 759) = 5.221$, $p = 0.006$. G: The burst amplitude, $F(2, 759) = 227.5$, $p < 0.001$. H: The probability of bursts to occur (illustrated as burst/s), $F(2, 27) = 6.335$, $p = 0.006$. Values are represented as mean + SEM. (i) Intraoperative and follow-up CT scans demonstrated the restoration of lateral ventricle morphology and the mechanical stability of the IVI, with no apparent displacement. (j) Immunofluorescence images showing the periventricular structures on the IVI-implanted and contralateral control sides of the sheep brain.

Quantitative analysis included the normalized cell density of NeuN-labelled neurons ($p = 0.22$), normalized cell density of Iba1-labelled microglial cells ($p = 0.65$) and normalized signal of GFAP-labelled astrocytes ($p = 0.83$). Two-tailed paired t tests were used.

References

1. Chen, C. et al. The Incidence and Risk Factors of Meningitis after Major Craniotomy in China: A Retrospective Cohort Study. *PLoS ONE* 9, e101961 (2014).
2. Lu, P. et al. Risk Factors of External Ventricular Drainage-Related Infections: A Retrospective Study of 147 Pediatric Post-tumor Resection Patients in a Single Center. *Front. Neurol.* 10, (2019).
3. Duy, P. Q. et al. Brain ventricles as windows into brain development and disease. *Neuron* 110, 12–15 (2022).
4. Fame, R. M., Cortés-Campos, C. & Sive, H. L. Brain Ventricular System and Cerebrospinal Fluid Development and Function: Light at the End of the Tube. *BioEssays* 42, 1900186 (2020).
5. Ge, Y.-J. et al. Genetic architectures of cerebral ventricles and their overlap with neuropsychiatric traits. *Nat. Hum. Behav.* 8, 164–180 (2024).
6. Grahn, J. A., Parkinson, J. A. & Owen, A. M. The cognitive functions of the caudate nucleus. *Prog. Neurobiol.* 86, 141–155 (2008).
7. Macpherson, T. & Hikida, T. Role of basal ganglia neurocircuitry in the pathology of psychiatric disorders. *Psychiat. Clin. Neuros.* 73, 289–301 (2019).
8. Banwinkler, M., Theis, H., Prange, S. & van Eimeren, T. Imaging the Limbic System in Parkinson's Disease—A Review of Limbic Pathology and Clinical Symptoms. *Brain Sci.* 12, 1248 (2022).
9. Khan, A. R. et al. Biomarkers of Parkinson's disease: Striatal sub-regional structural morphometry and diffusion MRI. *NeuroImage-Clin.* 21, 101597 (2019).
10. Munoz, F. et al. Long term study of motivational and cognitive effects of low-intensity focused ultrasound neuromodulation in the dorsal striatum of nonhuman primates. *Brain Stimul.* 15, 360–372 (2022).
11. Conde-Berriozabal, S. et al. Differential impact of optogenetic stimulation of direct and indirect pathways from dorsolateral and dorsomedial striatum on motor symptoms in Huntington's disease mice. *Exp. Neurol.* 383, 114991 (2025).
12. Jiang, S. et al. Spatially expandable fiber-based probes as a multifunctional deep brain interface. *Nat. Commun.* 11, 6115 (2020).
13. Zhang, S. et al. Body-Integrated, Enzyme-Triggered Degradable, Silk-Based Mechanical Sensors for Customized Health/Fitness Monitoring and In Situ Treatment. *Adv. Sci.* 7, 1903802 (2020).
14. Sun, L. et al. Implantable, Degradable, Therapeutic Terahertz Metamaterial Devices. *Small* 16, 2000294 (2020).
15. Zhou, Z. et al. The Use of Functionalized Silk Fibroin Films as a Platform for Optical Diffraction-Based Sensing Applications. *Adv. Mater.* 29, 1605471 (2017).
16. Shi, Z. et al. Silk-Enabled Conformal Multifunctional Bioelectronics for Investigation of Spatiotemporal Epileptiform Activities and Multimodal Neural Encoding/Decoding. *Adv. Sci.* 6, 1801617 (2019).
17. Lee, W. et al. A rewritable optical storage medium of silk proteins using near-field nano-optics.

- Nat. Nanotechnol.* **15**, 941–947 (2020).
18. Green, R. & Abidian, M. R. Conducting Polymers for Neural Prosthetic and Neural Interface Applications. *Adv. Mater.* **27**, 7620–7637 (2015).
 19. Hu, Z., Niu, Q., Hsiao, B. S., Yao, X. & Zhang, Y. Bioactive polymer-enabled conformal neural interface and its application strategies. *Mater. Horiz.* **10**, 808–828 (2023).
 20. Li, M. *et al.* Current status and outlook of biodegradable metals in neuroscience and their potential applications as cerebral vascular stent materials. *Bioact. Mater.* **11**, 140–153 (2022).
 21. Liu, K. *et al.* A Silk Cranial Fixation System for Neurosurgery. *Adv. Healthc. Mater.* **7**, 1701359 (2018).
 22. Liu, Y. *et al.* A high-density 1,024-channel probe for brain-wide recordings in non-human primates. *Nat. Neurosci.* **27**, 1620–1631 (2024).
 23. Murray, S. J. & Mitchell, N. L. The Translational Benefits of Sheep as Large Animal Models of Human Neurological Disorders. *Front. Vet. Sci.* **9**, (2022).
 24. Oxley, T. J. *et al.* Minimally invasive endovascular stent-electrode array for high-fidelity, chronic recordings of cortical neural activity. *Nat. Biotechnol.* **34**, 320–327 (2016).
 25. Beltrán, S. *et al.* Spinal cord motion and CSF flow in the cervical spine of 70 healthy participants. *NMR Biomed.* **37**, e5013 (2024).
 26. Wang, X. *et al.* Research progress on the osteogenic properties of tantalum in the field of medical implant materials. *J. Mater. Res. Technol.* **30**, 1706–1715 (2024).
 27. Zhao, Z. *et al.* Ultraflexible electrode arrays for months-long high-density electrophysiological mapping of thousands of neurons in rodents. *Nat. Biomed. Eng.* **7**, 520–532 (2023).
 28. Zhou, Y. *et al.* A silk-based self-adaptive flexible opto-electro neural probe. *Microsyst. Nanoeng.* **8**, 1–12 (2022).
 29. Chen, J. C. *et al.* Endocisternal interfaces for minimally invasive neural stimulation and recording of the brain and spinal cord. *Nat. Biomed. Eng.* 1–9 (2024) doi:10.1038/s41551-024-01281-9.

Point by point response (comments in black and responses in blue):

Reviewer #1:

The authors have fully addressed all the questions. I recommend the publication of this manuscript.
We appreciate the reviewer's recommendation for publication.

Reviewer #2:

This significantly improved manuscript demonstrates the feasibility of an intraventricular recording device, chronically implanted in sheep in the context of a Parkinson's disease model. The demonstration of a beta-power peak in the electrophysiological signals that is responsive to levodopa suggests that such an approach can in principle detect relevant disease biomarkers. While the extent of the "addressable" brain remains relatively limited (subcortical structures abutting the ventricle with sufficient ventricular volume locally to allow the device to unfold), and while there are no current indications for such an interface (even in Parkinson's, where activity in the non-ventricular-facing putamen rather than caudate is more relevant to motor symptoms that are currently treatable with deep brain stimulation), this is nonetheless a creative and interesting approach to expand the range of options available for interfacing with various brain structures, even if it is, at the current moment, a solution in search of a problem.

We sincerely thank the reviewer for recognizing the feasibility of our intraventricular interface and the significance of demonstrating a levodopa-responsive beta-power peak in the caudate nucleus. We acknowledge the current spatial limitations of "addressable" brain regions to subcortical structures adjacent to ventricles, and concur that immediate clinical translation requires further development. Nevertheless, this flexible, self-unfolding interface constitutes a foundational engineering advance enabling unprecedented chronic stability in deep-brain recordings (validated over one-month implantation), accessing neural populations beyond the reach of conventional technologies.

Beyond Parkinsonian models, this platform uniquely facilitates investigations into subcortical-cortical network dynamics, pharmacologically modulated oscillatory activities, and circuit-level pathophysiology—addressing critical gaps in existing methodologies. We have explicitly articulated these dimensions in the revised Discussion, emphasizing both the transformative potential of this approach and its present constraints while contextualizing its role in expanding the neurotechnology toolkit.